# SCENE TRANSFORMER: A UNIFIED ARCHITECTURE FOR PREDICTING MULTIPLE AGENT TRAJECTORIES

**Jiquan Ngiam**[*,1]**, Benjamin Caine**[*,1]**, Vijay Vasudevan**[*,1]**,**
**Zhengdong Zhang**[1]**, Hao-Tien Lewis Chiang**[2]**, Jeffrey Ling**[2]**,**
**Rebecca Roelofs**[1]**, Alex Bewley**[1]**, Chenxi Liu**[2]**, Ashish Venugopal**[2]**,**
**David Weiss**[2]**, Ben Sapp**[2]**, Zhifeng Chen**[1]**, Jonathon Shlens**[1]
[1]Google Research, Brain Team,[2] Waymo
{jngiam,bencaine,vrv}@google.com

## ABSTRACT

Predicting the motion of multiple agents is necessary for planning in dynamic environments. This task is challenging for autonomous driving since agents (e.g., vehicles and pedestrians) and their associated behaviors may be diverse and influence one another. Most prior work have focused on predicting independent futures for each agent based on all past motion, and planning against these independent predictions. However, planning against independent predictions can make it challenging to represent the future interaction possibilities between different agents, leading to sub-optimal planning. In this work, we formulate a model for predicting the behavior of all agents jointly, producing consistent futures that account for interactions between agents. Inspired by recent language modeling approaches, we use a masking strategy as the query to our model, enabling one to invoke a single model to predict agent behavior in many ways, such as potentially conditioned on the goal or full future trajectory of the autonomous vehicle or the behavior of other agents in the environment. Our model architecture employs attention to combine features across road elements, agent interactions, and time steps. We evaluate our approach on autonomous driving datasets for both marginal and joint motion prediction, and achieve state of the art performance across two popular datasets. Through combining a scene-centric approach, agent permutation equivariant model, and a sequence masking strategy, we show that our model can unify a variety of motion prediction tasks from joint motion predictions to conditioned prediction.

## 1 INTRODUCTION

Motion planning in a dense real-world urban environment is a mission-critical problem for deploying autonomous driving technology. Autonomous driving is traditionally considered too difficult for a single end-to-end learned system (Thrun et al., 2006). Thus, researchers have opted to split the task into sequential sub-tasks (Zeng et al., 2019): (i) perception, (ii) motion prediction, and (iii) planning. Perception is the task of detecting and tracking objects in the scene from sensors such as LiDARs and cameras. Motion prediction involves predicting the futures actions of other agents in the scene. Finally, planning involves creating a motion plan that navigates through dynamic environments.

Dividing the larger problem into sub-tasks achieves optimal performance when each sub-task is truly independent. However, such a strategy breaks down when the assumption of independence does not hold. For instance, the sub-tasks of motion prediction and planning are not truly independent—the autonomous vehicle's actions may significantly impact the behaviors of other agents. Similarly, the behaviors of other agents may dramatically change what is a good plan. The goal of this work is to take a step in the direction of unifying motion prediction and planning by developing a model that can exploit varying forms of conditioning information, such as the AV's goal, and produce joint consistent predictions about the future for all agents simultaneously.

While the motion prediction task has traditionally been formulated around per-agent independent predictions, recent datasets (Ettinger et al., 2021; Zhan et al., 2019) have introduced interaction prediction tasks that enable us to study joint future prediction (Figure 1). These interaction prediction

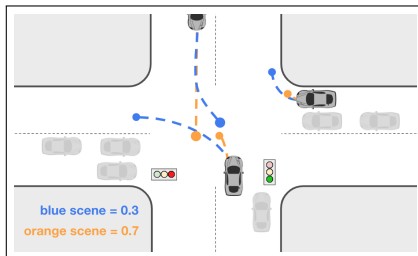

Figure 1: **Joint prediction provides consistent motion prediction.** Illustration of differences between marginal and joint motion prediction. Each color represents a distinct prediction. Left: Marginal prediction for bottom center vehicle. Scores indicate likelihood of trajectory. Note that the prediction is independent of other vehicle trajectories. Right: Joint prediction for three vehicles of interest. Scores indicate likelihood of entire scene consisting of trajectories of all three vehicles.

tasks require models to predict the joint futures of multiple agents: models are expected to produce future predictions for all agents such that the agents futures are consistent [1] with one another.

A naive approach to producing joint futures is to consider the exponential number of combinations of *marginal* agent predictions. Many of the combinations are not consistent, especially when agents have overlapping trajectories. We present a unified model that naturally captures the interactions between agents, and can be trained as a joint model to produce scene-level consistent predictions across all agents (Figure 1, right). Our model uses a scene-centric representation for all agents (Lee et al., 2017; Hong et al., 2019; Casas et al., 2020a; Salzmann et al., 2020) to allow scaling to large numbers of agents in dense environments. We employ a simple variant of self-attention (Vaswani et al., 2017) in which the attention mechanism is efficiently factorized across the agent-time axes. The resulting architecture simply alternates attention between dimensions representing time and agents across the scene, resulting in a computationally-efficient, uniform, and scalable architecture.

We find that the resulting model, termed *Scene Transformer*, achieves superior performance on both independent (marginal) and interactive (joint) prediction benchmarks.

Moreover, we demonstrate a novel formulation of the task using a masked sequence model, inspired by recent advances in language modeling (Brown et al., 2020; Devlin et al., 2019), to allow conditioning of the model on the autonomous vehicle (AV) goal state or full trajectory. In this reformulation, a single model can naturally perform tasks such as motion prediction, conditional motion prediction, and goal-conditioned prediction simply by changing which data is visible at inference time.

We hope that our unified architecture and flexible problem formulation opens up new research directions for further combining motion prediction and planning. In summary, our key contributions in this work are:

- A novel, scene-centric approach that allows us to gracefully switch training the model to produce either marginal (independent) and joint agent predictions in a single feed-forward pass. Our model achieves state-of-the-art on both marginal and joint prediction tasks on both the Argoverse and the Waymo Open Motion Dataset.

- A permutation equivariant Transformer-based architecture factored over agents, time, and road graph elements that exploits the inherent symmetries of the problem. The resulting architecture is efficient and integrates the world state in a unified way.

- A masked sequence modeling approach that enables us to condition on hypothetical agent futures at inference time, enabling conditional motion prediction or goal conditioned prediction.

## 2 RELATED WORK

**Motion prediction architectures.** Motion prediction models have flourished in recent years, due to the rise in interest in self-driving applications and the release of related datasets and bench-

---

[1]Marginal agent predictions may conflict with each other (have overlaps), while consistent joint predictions should have predictions where agents respect each other's behaviors (avoid overlaps) within the same future.

marks (Kesten et al., 2019; Chang et al., 2019; Caesar et al., 2020; Ettinger et al., 2021). Successful models must take into account the history of agent motion, and the elements of the road graph (e.g., lanes, stop lines, traffic light dynamic state). Furthermore, such models must learn the relationships between these agents in the context of the road graph environment.

One class of models draws heavily upon the computer vision literature, rendering inputs as a multi-channel rasterized top-down image (Cui et al., 2019; Chai et al., 2019; Lee et al., 2017; Hong et al., 2019; Casas et al., 2020a; Salzmann et al., 2020; Zhao et al., 2019). In this approach, relationships between scene elements are captured via convolutional deep architectures. However, the localized structure of the receptive field makes capturing spatially-distant interactions challenging. A popular alternative is to use an entity-centric approach. With this approach, agent state history is typically encoded via sequence modeling techniques like RNNs (Mercat et al., 2020; Khandelwal et al., 2020; Lee et al., 2017; Alahi et al., 2016; Rhinehart et al., 2019) or temporal convolutions (Liang et al., 2020). Road elements are approximated with basic primitives (e.g. piecewise-linear segments) which encode pose information and semantic type. Modeling relationships between entities is often presented as an information aggregation process, and models employ pooling (Zhao et al., 2020; Gao et al., 2020; Lee et al., 2017; Alahi et al., 2016; Gupta et al., 2018), soft-attention (Mercat et al., 2020; Zhao et al., 2020; Salzmann et al., 2020) as well as graph neural networks (Casas et al., 2020a; Liang et al., 2020; Khandelwal et al., 2020).

Like our proposed method, several recent models use Transformers (Vaswani et al., 2017), composed of multi-headed attention layers. Transformers are a popular state-of-the-art choice for sequence modeling in natural language processing (Brown et al., 2020; Devlin et al., 2019), and have recently shown promise in core computer vision tasks such as detection (Bello et al., 2019; Carion et al., 2020; Srinivas et al., 2021), tracking (Hung et al., 2020) and classification (Ramachandran et al., 2019; Vaswani et al., 2021; Dosovitskiy et al., 2021; Bello, 2013; Bello et al., 2019). For motion modeling, recent work has employed variations of self-attention and Transformers for modeling different axes: temporal trajectory encoding and decoding (Yu et al., 2020; Giuliari et al., 2020; Yuan et al., 2021), encoding relationships between agents (Li et al., 2020; Park et al., 2020; Yuan et al., 2021; Yu et al., 2020; Mercat et al., 2020; Bhat et al., 2020), and encoding relationships with road elements. When applying self-attention over multiple axes, past work used independent self-attention for each axis (Yu et al., 2020), or flattened two axes together into one joint self-attention layer (Yuan et al., 2021) – by comparison, our method proposes axis-factored attention to model relationships between time steps, agents, and road graph elements in a unified way.

**Scene-centric versus agent-centric representations.** Another key design choice is the frame of reference in which the representation is encoded. Some models do a majority of modeling in a global, scene-level coordinate frame, such as work that employs a rasterized top-down image (Cui et al., 2019; Chai et al., 2019; Lee et al., 2017; Hong et al., 2019; Casas et al., 2020a; Salzmann et al., 2020). This can lead to a more efficient model due to a single shared representation of world state in a common coordinate frame, but comes with the potential sacrifice of pose-invariance. On the other hand, models that reason in the agent-coordinate frame (Mercat et al., 2020; Zhao et al., 2020; Khandelwal et al., 2020) are intrinsically pose-invariant, but scale linearly with the number of agents, or quadratically with the number of pairwise interactions between agents. Many works employ a mix of a top-down raster representation for road representation fused with a per-agent representations (Rhinehart et al., 2019; Tang & Salakhutdinov, 2019; Lee et al., 2017). Similar to our own work, LaneGCN (Liang et al., 2020) is agent-centric yet representations are in a global frame – to the best of our knowledge, this is the only other work to do so. This enables efficient reasoning while capturing arbitrarily distant interactions and high-fidelity state representations without rasterization.

**Representing multi-agent futures.** A common way to represent agent futures is via a weighted set of trajectories per agent (Alahi et al., 2016; Biktairov et al., 2020; Buhet et al., 2020; Casas et al., 2020a;a; Chai et al., 2019; Cui et al., 2019; Gao et al., 2020; Hong et al., 2019; Lee et al., 2017; Marchetti et al., 2020; Mercat et al., 2020; Salzmann et al., 2020; Zhao et al., 2020; Chandra et al., 2018). This representation is encouraged by benchmarks which primarily focus on per-agent distance error metrics (Caesar et al., 2020; Chang et al., 2019; Zhan et al., 2019). We argue in this work that modeling *joint* futures in a multi-agent environment (Figure 1, right) is an important concept that has been minimally explored in prior work. Some prior work consider a factorized pairwise joint distribution, where a subset of agent futures are conditioned on other agents – informally, modeling $P(X)$ and $P(Y|X)$ for agents $X$ and $Y$ (Khandelwal et al., 2020; Tolstaya et al., 2021; Salzmann et al., 2020). To generalize joint prediction to arbitrary multi-agent settings, other work (Tang

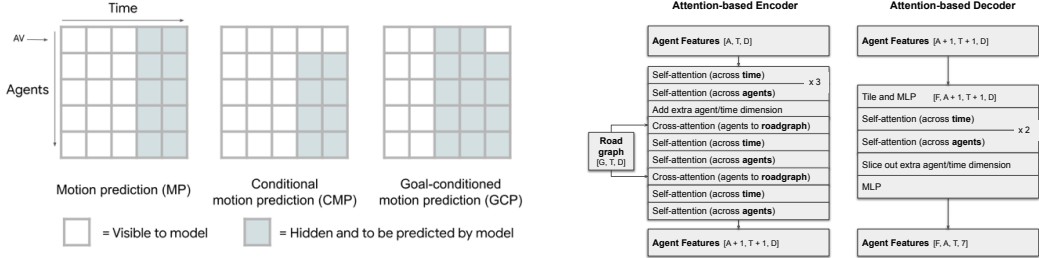

Figure 2: **Single model architecture for multiple motion prediction tasks.** Left: Different masking strategies define distinct tasks. The left column represents current time and the top row represents the agent indicating the autonomous vehicle (AV). A single model can be trained for data associated with motion prediction, conditional motion prediction, and goal-directed prediction, by matching the masking strategy to each prediction task. Right: Attention-based encoder-decoder architecture for joint scene modeling. Architecture employs factored attention along the time and agent axes to exploit the dependencies in the data, and cross-attention to inject side information.

& Salakhutdinov, 2019; Rhinehart et al., 2019; Casas et al., 2020b; Suo et al., 2021; Yeh et al., 2019) iteratively roll out samples per-agent, where each agent is conditioned on previously sampled trajectory steps. In contrast, our model directly decodes a set of $k$ distinct joint futures with associated likelihoods.

## 3  METHODS

The Scene Transformer model has three stages: (i) Embed the agents and the road graph into a high dimensional space, (ii) Employ an attention-based network to encode the interactions between agents and the road graph, (iii) Decode multiple futures using an attention-based network. The model takes as input a feature for every agent at every time step, and also predicts an output for every agent at every time step. We employ an associated mask, where every agent time step has an associated indicator of 1 (hidden) or 0 (visible), indicating whether the input feature is hidden (i.e. removed) from the model. This approach mirrors the approach of masked-language models such as BERT (Devlin et al., 2019). The approach is flexible, enabling us to simultaneously train a single model for *motion prediction (MP)* (Cui et al., 2019; Chai et al., 2019; Lee et al., 2017; Hong et al., 2019; Casas et al., 2020a; Salzmann et al., 2020; Casas et al., 2020a; Liang et al., 2020; Khandelwal et al., 2020), *conditional motion prediction (CMP)* (Khandelwal et al., 2020; Tolstaya et al., 2021; Salzmann et al., 2020) and *goal-conditioned prediction (GCP)* (Deo & Trivedi, 2020) simply by changing what data is shown to the model (Figure 2, left). We summarize the key contributions below, and reserve details for the Appendix.

**Multi-task representation.** The key representation in the model is a 3-dimensional tensor of $A$ agents with $D$ feature dimensions across $T$ time steps. At every layer within the architecture, we aim to maintain a representation of shape $[A, T, D]$, or when decoding, $[F, A, T, D]$ across $F$ potential futures. Each task (MP, CMP, GCP) can be formulated as a query with a specific masking strategy by setting the indicator mask to 0, thus providing that data to the model (Figure 2, left). The goal of the model is to impute the features for each shaded region corresponding to subsets of time and agents in the scenario that are masked.

### 3.1  SCENE-CENTRIC REPRESENTATION FOR AGENTS AND ROAD GRAPHS.

We use a scene-centric embedding where we use an agent of interest's position as the origin [2], and encode *all* roadgraph and agents with respect to it. This is contrast to approaches which use an agent-centric representation, where the representations are computed separately for each agent, treating each agent in turn as the origin.

---

[2]For WOMD, we center the scene with respect to the autonomous vehicle (AV). For Argoverse, we center the scene with respect to the agent that needs to be predicted. Both are centered around what would be the last visible time step in a motion prediction setup for all tasks.

In detail, we first generate a feature for every agent time step if that time step is visible. Second, we generate a set of features for the static road graph, road elements static in space and time, learning one feature vector *per polyline* (with signs being polylines of length 1) using a PointNet (Qi et al., 2017). Last, we generate a set of features for the dynamic road graph, which are road elements static in space but dynamic in time (e.g. traffic lights), also one feature vector per object. All three categories have $xyz$ position information, which we preprocess to center and rotate around the agent of interest and then encode with sinusoidal position embeddings (Vaswani et al., 2017).

### 3.2 ENCODING TRANSFORMER

We focus on a simple encoder-decoder attention-based architecture which maintains a representation of $[A, T, D]$ throughout (Figure 2, right). We summarize the architecture briefly, but reserve details for the Appendix and Table 5. The majority of layers are a form of the Transformer layer (Vaswani et al., 2017) (Table 6). Attention layers are parameterized as matrices representing the query $Q$, key $K$, and value $V$, whose output $y = \mathsf{softmax}\frac{(Q\,K^T)V}{\sqrt{dim_k}}$. Each matrix is computed as a learned linear transformation of the underlying representation $x$, e.g. $Q = W_q\,x$. Each attention layer is followed by a feed-forward layer of the same hidden dimension, and a skip connection addition of the result with the input to the whole Transformer layer. All layers of the encoder and decoder employ a $D$ feature dimension. The final layer after the decoder is a 2-layer MLP that predicts 7 outputs. The first 6 outputs correspond to the 3-dimensional position of an agent at a given time step in *absolute coordinates* (e.g. meters) with respect to the agent of interest, and the corresponding uncertainty parameterized by a Laplace distribution (Meyer & Thakurdesai, 2020). The remaining dimension predicts the heading.

**Efficient factorized self-attention.** The bulk of the computation is performed with a Transformer (Vaswani et al., 2017) (Table 6). One naive approach to use the Transformer would be to perform attention directly on the entire set of agent and time step features (i.e., attention across $AT$ dimensions). However, this approach is computationally expensive, and also suffers from an identity symmetry challenge: since we do not add any specific agent identity indicator, two agents of the same type with the same masked future time-step will have the same input representation to the transformer, resulting in the same output. Thus, we design a *factorized* attention based on the time and agents axes (for related ideas, see Wang et al. (2020); Szegedy et al. (2016); Ho et al. (2019)).

Applying attention only across time allows the model to learn smooth trajectories independent of the identity of the agent. Likewise, applying attention only across agents allows the model to learn multi-agent interactions independent of the specific time step. Finally, in order to capture both time and agent dependencies, the model simply alternates attention across agents and time in subsequent layers (Figure 2, right panel). The model is also permutation equivariant to the ordering of the agents at input, since the attention operation is permutation equivariant.

**Cross-attention.** In order to exploit side information, which in our case is a road graph, we use cross-attention to enable the agent features to be updated by attending to the road graph. Concretely, we calculate the queries from the agents, but the keys and values come from the embeddings of the road graph. The road graph embeddings are final after the per-polyline PointNet, and therefore *not* updated during these attention layers. This requires that the model learn interactions between the road structure and agents that are independent of the specific time step or agent. We highlight that the road graph representation is also permutation-equivariant and shared across all agents in the scene, whereas prior approaches have often used a per-agent road graph representation.

### 3.3 PREDICTING PROBABILITIES FOR EACH FUTURES.

Our model also needs to predict a probability score for each future (in the joint model) or trajectory (in the marginal model). In order to do so, we need a feature representation that summarizes the scene and each agent. After the first set of factorized self-attention layers, we compute the mean of the agent features tensor across the agent and time dimension separately, and add these as an additional artificial agent and time making our internal representation $[A + 1, T + 1, D]$ (Figure 2, left panel). This artificial agent and time step propagates through the network, and provides the model with extra capacity for representing each agent, that is not tied to any timestep. At the final layer, we slice out the artificial agent and time step to obtain summary features for each agent (the additional time per

agent), and for the scene (the 'corner' feature that is both additional time and agent). This feature is then processed by a 2-layer MLP producing a single logit value that we use with a softmax classifier for a permutation equivariant estimate of probabilities for each futures.

### 3.4 JOINT AND MARGINAL LOSS FORMULATION.

The output of our model is a tensor of shape $[F, A, T, 7]$ representing the location and heading of each agent at the given time step. Because the model uses a scene-centric representation for the locations through positional embeddings, the model is able to predict all agents simultaneously in a single feed-forward pass. This design also makes it possible to have a straight-forward switch between joint future predictions and marginal future predictions.

To perform joint future prediction, we treat each future (in the first dimension) to be coherent futures across all agents. Thus, we aggregate the displacement loss across all agents [3] and time steps to build a loss tensor of shape $[F]$. We only back-propagate the loss through the individual future that most closely matches the ground-truth in terms of displacement loss (Gupta et al., 2018; Yeh et al., 2019). For marginal future predictions, each agent is treated independently. After computing the displacement loss of shape $[F, A]$, we do *not* aggregate across agents. Instead, we select the future with minimum loss for each agent separately, and back-propagate the error correspondingly (Appendix, Figure 8).

**Evaluation metrics for motion prediction.** We evaluate the quality of $k$ weighted trajectory hypotheses using the standard evaluation metrics: minADE, minFDE, miss rate, and mAP. Each evaluation metric attempts to measure how close the top $k$ trajectories are to ground truth observation. A simple and common distance-based metric is to measure the $L_2$ norm between a given trajectory and the ground truth (Alahi et al., 2016; Pellegrini et al., 2009). minADE reports the $L_2$ norm of the trajectory with the minimal distance. minFDE likewise reports the $L_2$ norm of the trajectory with the smallest distance only evaluated at the final location of the trajectory. We additionally report the miss rate (MR) and mean average precision (mAP) to capture how well a model predicts all of the future trajectories of agents probabilistically (Yeh et al., 2019; Chang et al., 2019; Ettinger et al., 2021). For joint future evaluation settings, we measure the scene-level equivalents (minSADE, minSFDE, and SMR) that evaluate the prediction of the best single consistent future (Casas et al., 2020b).

## 4 RESULTS

We evaluate the Scene Transformer on motion prediction tasks from the Argoverse dataset (Chang et al., 2019) and Waymo Open Motion Dataset (WOMD) (Ettinger et al., 2021). The Argoverse dataset consists of 324,000 run segments (each 5 seconds in length) from 290 km of roadway containing 11.7 million agent tracks in total, and focuses on single agent (marginal) motion prediction. The WOMD dataset consists of 104,000 run segments (each 20 seconds in length) from 1,750 km of roadway containing 7.64 million unique agent tracks. Importantly, the WOMD has two tasks, each with their own set of evaluation metrics: a *marginal motion prediction challenge* that evaluates the quality of the motion predictions independently for each agent (up to 8 per scene), and a *joint motion prediction challenge* that evaluates the quality of a model's joint predictions of exactly 2 agents per scene. We train each model on a Cloud TPU (Jouppi et al., 2017) See Appendix for all training details.

First, in Section 4.1, we focus on the marginal prediction task and show that Scene Transformer achieves competitive results on both Argoverse (Chang et al., 2019) and the WOMD (Ettinger et al., 2021). In Section 4.2, we focus on the joint prediction task and train the Scene Transformer with our joint loss formulation (see Section 3.4). We show that with a single switch of the loss formula, we can achieve superior *joint motion prediction* performance. In Section 4.3 we discuss factorized versus non-factorized attention. Finally, in Section 4.4 we show how our masked sequence model formulation allows us to train a multi-task model capable of motion prediction, conditional motion prediction, and goal-conditioned prediction. In Appendix B.1 we discuss the trade-off between marginal and joint models.

---

[3]Each dataset identifies a subset of agents to be predicted. We only include this subset in our loss calculation. For Argoverse, this is 1 agent; for WOMD this is 2-8 agents.

| Method | minADE↓ | minFDE↓ | MR↓ |
|---|---|---|---|
| Jean (Mercat et al., 2020) | 0.97 | 1.42 | 0.13 |
| WIMP (Khandelwal et al., 2020) | 0.90 | 1.42 | 0.17 |
| TNT (Zhao et al., 2020) | 0.94 | 1.54 | 0.13 |
| LaneGCN (Liang et al., 2020) | 0.87 | 1.36 | 0.16 |
| TPCN (Ye et al., 2021) | 0.85 | 1.35 | 0.16 |
| mmTransformer (Liu et al., 2021b) | 0.84 | 1.34 | 0.15 |
| HOME (Gilles et al., 2021) | 0.94 | 1.45 | **0.10** |
| Ours (marginal) | **0.80** | **1.23** | 0.13 |

Table 1: **Marginal predictive performance on Argoverse motion prediction.** Results reported on *test* split for vehicles (Chang et al., 2019). minADE, minFDE reported for $k = 6$ predictions (Alahi et al., 2016; Pellegrini et al., 2009); Miss Rate (MR) (Chang et al., 2019) within 2 meters of the target. All results are reported for $t = 3$ seconds.

| Motion Prediction | minADE↓ | | | minFDE↓ | | | MR↓ | | | mAP↑ | | |
|---|---|---|---|---|---|---|---|---|---|---|---|---|
| | veh | ped | cyc | veh | ped | cyc | veh | ped | cyc | veh | ped | cyc |
| *valid* | | | | | | | | | | | | |
| LSTM baseline (Ettinger et al., 2021) | 1.34 | 0.63 | 1.26 | 2.85 | 1.35 | 2.68 | 0.25 | 0.13 | 0.29 | 0.23 | 0.23 | 0.20 |
| Ours (marginal) | **1.17** | **0.59** | **1.15** | **2.51** | **1.26** | **2.44** | **0.20** | **0.12** | **0.24** | **0.26** | **0.27** | **0.20** |
| *test* | | | | | | | | | | | | |
| LSTM baseline (Ettinger et al., 2021) | 1.34 | 0.64 | 1.29 | 2.83 | 1.35 | 2.68 | 0.24 | 0.13 | 0.29 | 0.24 | 0.22 | 0.19 |
| ReCoAt (Huang et al., 2021) | 1.69 | 0.69 | 1.47 | 3.96 | 1.51 | 3.30 | 0.40 | 0.20 | 0.37 | 0.18 | 0.25 | 0.17 |
| SimpleCNNOnRaster (Konev et al., 2021) | 1.47 | 0.71 | 1.39 | 3.18 | 1.52 | 2.89 | 0.27 | 0.16 | 0.31 | 0.19 | 0.18 | 0.14 |
| DenseTNT (Gu et al., 2021) | 1.35 | 0.85 | 2.17 | 3.35 | 1.40 | 2.94 | 0.20 | 0.13 | 0.23 | **0.28** | **0.28** | **0.21** |
| Ours (marginal) | **1.17** | **0.60** | **1.17** | **2.48** | **1.25** | **2.43** | **0.19** | **0.12** | **0.22** | 0.27 | 0.23 | 0.20 |

Table 2: **Marginal predictive performance on Waymo Open Motion Dataset motion prediction.** Results presented on the *standard* splits of the validation and test datasets (Ettinger et al., 2021) evaluated with traditional *marginal metrics* for $t = 8$ seconds. minADE, minFDE reported for $k = 6$ predictions (Alahi et al., 2016; Pellegrini et al., 2009); Miss Rate (MR) (Chang et al., 2019) within 2 meters of the target. See Appendix for $t = 3$ or 5 seconds. We include the challenge winner results in this table (Waymo, 2021).

## 4.1 MARGINAL MOTION PREDICTION

We first evaluate the performance of Scene Transformer trained and evaluated as a traditional marginal, per-agent motion prediction model. This is analogous to the problem illustrated in Figure 1 (left). For all results until Section 4.4, we use a masking strategy that provides the model with all agents as input, but with their futures hidden. We also mask out future traffic light information.

**Argoverse.** We evaluate on the popular Argoverse (Chang et al., 2019) benchmark to demonstrate the efficacy of our architecture. During training and evaluation, the model is only required to predict the future of the single agent of interest. Our best Argoverse model uses $D = 512$ feature dimensions and label smoothing for trajectory classification. Our model achieves state-of-the-art results compared to published, prior work [4] in terms of minADE and minFDE (Table 1).

**Waymo Open Motion Dataset (WOMD).** We next evaluate the performance of our model with $D = 256$ on the recently released WOMD (Ettinger et al., 2021) for the marginal motion prediction task. This task is a standard motion prediction task where up to 8 agents per scene are selected to have their top 6 motion predictions evaluated independently. Our model trained with the marginal loss achieves state-of-the-art results on the minADE, minFDE, and miss rate metrics (Table 2).

## 4.2 JOINT MOTION PREDICTION

To evaluate the effectiveness of Scene Transformer when trained with a joint loss formulation (Section 3.4), we evaluate our model on the *Interaction Prediction* challenge in WOMD (Ettinger et al., 2021). This task measures the performance of the model at predicting the *joint* future trajectories of two

---

[4]We exclude comparing to public leaderboard entries that have not been published since their details are not available, but note that our results are competitive on the leaderboard as of the submission date.

| Interaction Prediction | minSADE↓ | | | minSFDE↓ | | | SMR↓ | | | mAP↑ | | |
|---|---|---|---|---|---|---|---|---|---|---|---|---|
| | veh | ped | cyc | veh | ped | cyc | veh | ped | cyc | veh | ped | cyc |
| *valid* | | | | | | | | | | | | |
| LSTM baseline (Ettinger et al., 2021) | 2.42 | 2.73 | 3.16 | 6.07 | 4.20 | 6.46 | 0.66 | 1.00 | 0.83 | 0.07 | **0.06** | 0.02 |
| Ours (marginal-as-joint) | 2.04 | 1.62 | 2.28 | 4.94 | 3.81 | 5.67 | 0.54 | 0.63 | 0.72 | **0.11** | 0.05 | 0.03 |
| Ours (joint, MP-only) | **1.72** | **1.38** | 1.96 | **3.98** | **3.11** | 4.75 | **0.49** | **0.60** | 0.73 | **0.11** | 0.05 | 0.03 |
| Ours (joint, multi-task) | **1.72** | 1.39 | **1.94** | 3.99 | 3.15 | **4.69** | **0.49** | 0.62 | **0.71** | **0.11** | **0.06** | **0.04** |
| *test* | | | | | | | | | | | | |
| LSTM baseline (Ettinger et al., 2021) | 2.46 | 2.47 | 2.96 | 6.22 | 4.30 | 6.26 | 0.67 | 0.89 | 0.89 | 0.06 | 0.03 | 0.03 |
| HeatIRm4 (Mo et al., 2021) | 2.93 | 1.77 | 2.74 | 7.20 | 4.06 | 6.69 | 0.80 | 0.80 | 0.91 | 0.07 | **0.05** | 0.00 |
| Ours (marginal-as-joint) | 2.08 | 1.62 | 2.24 | 5.04 | 3.87 | 5.41 | 0.55 | 0.64 | 0.73 | 0.08 | **0.05** | 0.03 |
| Ours (joint, MP-only) | 1.76 | **1.38** | 1.95 | 4.08 | **3.19** | 4.65 | **0.50** | 0.62 | **0.70** | 0.10 | **0.05** | **0.04** |
| Ours (joint, multi-task) | **1.74** | 1.41 | **1.95** | **4.06** | 3.26 | 4.68 | **0.50** | 0.64 | 0.71 | **0.13** | 0.04 | 0.03 |

Table 3: **Joint predictive performance on Waymo Open Motion Dataset motion prediction**. Results presented on the *interactive* splits of the validation and test datasets (Ettinger et al., 2021) evaluated with scene-level *joint metrics* for $t = 8$ seconds. minSADE, minSFDE for $k = 6$ predictions (Alahi et al., 2016; Pellegrini et al., 2009); Miss Rate (MR) (Chang et al., 2019) within 2 meters of the target. "**S**" indicates a scene-level joint metric. See Appendix for $t = 3$ or 5 seconds. We include the challenge winner results in this table (Waymo, 2021).

interacting agents (Figure 1, right), and employs *joint variants* of the common minADE, minFDE, and Miss Rate (MR) metrics denoted as minSADE, minSFDE, SMR. Note that the "**S**" indicates "scene-level" metrics. These metrics aim to measure the quality and consistency of the two agents joint prediction - for example, the joint variant of Miss Rate (SMR) only records a "hit" if both interacting agent's predicted trajectories are within the threshold of their respective ground truths.

We find that for the *Interaction Prediction* challenge our joint model's joint predictions easily outperforms the WOMD provided baseline as well as a marginal version of the model converted into a joint prediction [5] into joint predictions. (Table 3). This shows that beyond the strength of our overall architecture and approach, that explicitly training a model as a joint model significantly improves joint performance on joint metrics. A notable observation is that even though the *Interaction Prediction* task only requires predicting the joint trajectories of two agents, our method is fully general and predicts joint consistent futures of all agents.

### 4.3 Factorized agents self-attention

Factorized self-attention confers two benefits to our model: (a) it is more efficient since the attention is over a smaller set, and (b) it provides an implicit identity to each agent during the attention across time. We ran an experiment where we replaced each axis-factorized attention layer (each pair of time and agent factorized layer) with a non-axis factored attention layer. This increased the computational cost of the model and performed worse on the Argoverse validation dataset: the factorized version achieved a minADE of 0.609 with the factored version, and 0.639 with the non-factorized version.

### 4.4 Advantages of a masked sequence modeling strategy

Our model is formulated as a masked sequence model (Devlin et al., 2019), where at training and inference time we specify which agent timesteps to mask from the model. This formulation allows us to select which information about any agent at any timestep to supply to the model, and measure how the model exploits or responds to this additional information. We can express several motion prediction tasks at inference time in mask space (Figure 2), in effect providing a multi-task model. We can use this unique capability to query the models for counterfactuals. What would the model predict given a subset of agents full trajectories (conditional motion prediction), or given a subset of agents final goals (goal-conditioned motion prediction)? This feature could be particularly useful for autonomous vehicle planning to predict what various future rollouts of the scene would look like given a desired goal for the autonomous vehicle (Figure 3).

---

[5]Note that the output of a joint model can be directly used in a marginal evaluation. However, converting the output of a marginal model into a joint evaluation is nuanced because there lacks an association of futures across agents (Figure 1). We employ a simple heuristic to convert the outputs of a marginal model for joint evaluation: we take the top 6 pairs of trajectories from the combination of both agent's trajectories for 36 total pairs, and retain the top 6 pairs with the highest product of probabilities.

*Scenario A*          *Scenario B*          *Scenario C*

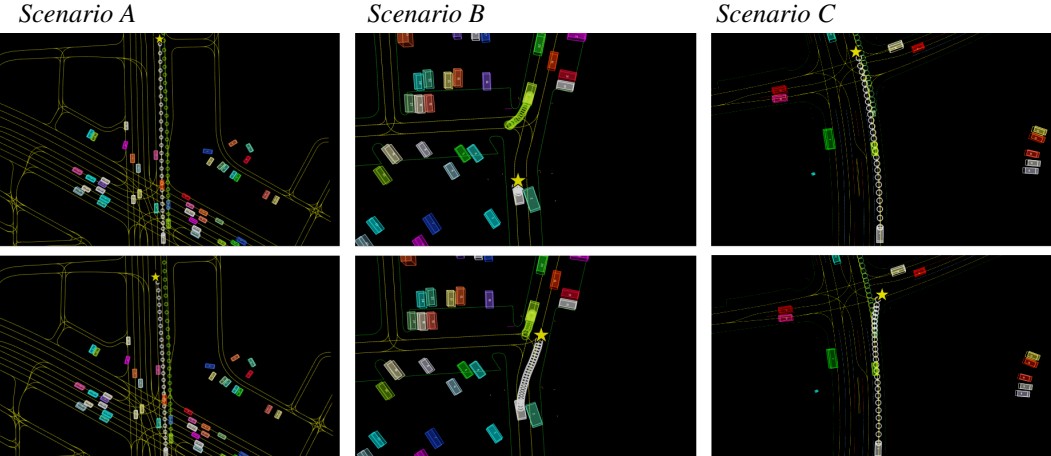

Figure 3: **Goal-conditioned prediction navigates AV to selected goal positions.** Rectangles indicate vehicles on the road. Lines indicate the road graph (RG) colored by the type of lane marker. Circles indicate predicted trajectory for each agent. Star indicates selected goal for AV. Each column represents a scenario in which the AV is directed to perform one of two actions taking into account the dynamics of other agents. (A) AV instructed to either change lanes or remain in the same lane. (B) AV instructed to stop to allow oncoming vehicle to turn left, or adjust to the side in a narrow passage so that the oncoming vehicle has enough space to pass. (C) AV instructed to either proceed straight or turn right at the intersection.

In the previous results, we used a prefix-mask during training and inference, which shows the first few steps and predicts the remaining timesteps for all agents. For this experiment, we employ two masking strategies to test on the interactive split of the WOMD, namely "conditional motion prediction" (CMP) where we show one of the two interacting agents' full trajectory, and "goal-conditioned motion prediction" (GCP) where we show the autonomous vehicle's desired goal state (Figure 2). We train a model using each strategy (including MP masking) $1/3$ of the time, and evaluate the model on each all tasks. We find that across the 44k evaluation segments, the multi-task model matches the performance of our MP-only trained joint model on both joint (Table 3) and marginal metrics (see also Appendix, Table 11); multi-tasking training does not significantly degrade standard motion prediction performance. We qualitatively examine the performance of the multi-task model in a GCP setting, and observe that the joint motion predictions for the AV and non-AV agents flexibly adapt to selected goal points for the AV (Figure 3).

We note that dataset average metrics like min(S)ADE do not capture forecasting subtleties (Ivanovic & Pavone, 2021) that are critical for measuring progress in the field. Unfortunately, the Waymo Motion Open Dataset Interactive Prediction set does not contain pre-defined metadata necessary to evaluate specific interesting slices using different models. In Appendix C we provide some initial analysis of slices that we could compute on the dataset, which illustrate how joint and marginal model predictions behavior differently as the scene a) scales with more agents in complicated scenes, b) differs in the average speed of agents. We also show that joint prediction models demonstrate *joint consistent futures* via lower inter-prediction overlap compared to marginal models.

## 5 DISCUSSION

We propose a unified architecture for autonomous driving that is able to model the complex interactions of agents in the environment. Our approach enables a single model to perform motion prediction, conditional motion prediction, and goal-conditioned prediction. In the field of autonomous driving, elaborations of this problem formulation may result in learning models for planning systems that quantitatively improve upon existing systems (Buehler et al., 2009; Montemerlo et al., 2008; Ziegler et al., 2014; Zeng et al., 2019; Liu et al., 2021a). Likewise, such modeling efforts may be used to directly go after issues of identifying interacting agents in an environment, and potentially provide an important task for identifying causal relationships (Arjovsky et al., 2019; Schölkopf et al., 2021).

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

# Appendix

## A    ARCHITECTURE AND TRAINING DETAILS

**Overview.** Table 5 provides additional details about the Scene Transformer architecture and training. The bulk of the computation and parameters reside in the Transformer layers. Table 6 lists all of the operations and learned parameters in our implementation of a Transformer. MLPs are employed to embed the original data (e.g, layers $A$, $B$, $C$), build $F$ futures from the encoding (layer $T$), and predict the final position, headings and uncertainties of all agents (layers $Z_1$ and $Z_2$). The resulting attention-based architecture using $D = 256$ feature dimensions contains 15,296,136 parameters. Nearly all settings used for both WOMD and Argoverse are identical, except that for Argoverse we use $D = 512$ and label smoothing to get our best marginal results. The resulting model is trained on a TPU custom hardware accelerator (Jouppi et al., 2017) and converges in about 3 days of training.

**Decoding multiple futures.** In order to allow the decoder to output $F$ distinct futures, we perform the following operations. The decoder receives as input a tensor of shape $[A, T, D]$ corresponding to $A$ agents across $T$ time steps and $D$ feature dimensions. The following series of simple operations restructures the representation to predict $F$ futures. First, the representation is tiled $F$ times to generate $[F, A, T, D]$. We append a one-hot encoding to the final dimension where a 1 indicates which the identity of each future, resulting in a tensor of shape $[F, A, T, D + F]$. The one-hot encoding allows the network to learn an embedding for each of the $F$ futures. For computational simplicity, the resulting representation is propagated through a small MLP to produce a return a tensor of the shape $[F, A, T, D]$.

**Padding and Hidden Masks.** Padding and hidden masks are important to get right in the implementation of such a model. In particular, we need to ensure that the masks do not convey additional future information to the model (e.g., if the model knows which timesteps an agent is visible or occluded based on the padding in the data, it may take advantage of this and not be able to generalize). We use the concept of padding to indicate the positions where the *input* is absent or provided. This is distinct from the hidden mask of shape $[A, T]$ that is used for *task specification*. The hidden mask is used to query the model to inform it on which locations to predict, while the padding tells us which locations have inputs and ground-truth to compute a loss over. All padded positions are set to be hidden during preprocessing, regardless of the masking scheme, so the model tries to predict their values. All layers in our implementation are padding aware, including normalization layers (like batch normalization used in our encoding MLPs) and attention operations. Our attention operations set the attention weights to 0 for padded elements. If after providing our task specific hidden mask, no non-padded (valid) time steps exist for an agent, the whole agent is set to be padded. This prevents agent slots with no valid data from being used in the model.

**Predicting uncertainties.** For each predicted variate such as position or heading, we additionally predict a corresponding uncertainty. In early experiments we found that predicting a paired uncertainty improves the predictive performance of the original variate and provides a useful signal for interpreting the fidelity of the predicted value. We employ a loss function that predicts a parameterization of the uncertainty corresponding to a Laplace distribution (Meyer & Thakurdesai, 2020).

**Loss in lateral and longitudinal coordinates.** We use a Laplace distribution parameterization for the loss, with a diagonal covariance such that each axis is independent. To enable the model to learn meaningful uncertainties, we rotate the scene per box prediction such that each prediction's associated ground-truth is axis aligned. This formulation results in uncertainty estimates that correspond to the agent's lateral and longitudinal coordinates.

**Heading representation:** Our output format is 7 dimensional, where one of the dimensions, heading, is represented in radians. To supervise the loss on the heading dimension, we employ a now standard method of "wrapping" the angle difference between the predicted and groundtruth heading for each agent; see Figure 4.

**Marginal predictions from a joint model.** The model has the added feature of being easily adapted for making marginal predictions for each agent. To produce per-trajectory scores for each agents predictions, we attach an artificial extra time step to the end of the $[A, T, D]$ agent feature matrix to give us a matrix of shape $[A, T + 1, D]$. This additional feature provides the model with extra

```
def WrapAngleRadians(angles_rad, min_val=-np.pi, max_val=np.pi):
  max_min_diff = max_val - min_val
  return min_val + tf.math.floormod(angles_rad + max_val, max_min_diff)

heading_diff = pred_heading - gt_heading
heading_error = WrapAngleRadians(heading_diff, min_val=-PI, max_val=PI)
heading_loss = HuberLoss(heading_error)
```

Figure 4: Pseudo-code for heading loss component.

| With Augmentation? | Vehicle minADE @8s | Ped minADE @8s | Cyclist minADE @ 8s |
|---|---|---|---|
| No | 1.30 | 0.69 | 1.29 |
| Yes | 1.18 | 0.59 | 1.15 |

Table 4: WOMD validation set performance with and without data augmentation during training.

capacity for representing each agent, that is not tied to any timestep. This additional feature can be used to predict which of the $k$ trajectories most closely matches the ground truth for that agent.

**Data augmentation.** We use two methods for data augmentation to aid generalization and combat overfitting on both datasets. First, we use agent dropout, where we artificially remove non-predicted agents with some probability. We found a probability of $0.1$ worked well for both datasets. We also found it beneficial to randomly rotate the entire scene between $[-\frac{\pi}{2}, \frac{\pi}{2}]$ after centering to the agent of interest. On Argoverse, this agent of interest is the agent the task designates to be predicted, where as on the WOMD we always center around the autonomous vehicle (AV), even though for some tasks it is not one of the predicted agents. Lastly, each Argoverse scene contains many agents that the model is not required to predict; we employ these contextual agents as additional target training data if the contextual agents moved by at least 6m. On WOMD, Table 4 shows data augmentation on the standard validation set does improve minADE modestly, though does not account for the majority of the benefit of the model.

**Argoverse classification label smoothing:** Our best model on Argoverse uses $D = 512$ feature dimensions, but naively scaling the model this way leads to severe overfitting on the classification subtask. During training of our best model we employ label smoothing (0.1 + 1/6 for negatives and 0.9 + 1/6 for positive target).

**WOMD redundant trajectory combination:** The AP evaluation metric on the WOMD expects that exactly one of the trajectories is given a high probability and the other trajectories a low probability. For example, the future prediction for a stationary agent is expected to have only one future with a high probability score, with a low score for the rest – even though the trajectories may all be the same, in this case, stationary. Our best model combines redundant trajectories together if they are close in spatial location (less than 3.2m) at the final timestep.

**Embedding of agents and road graph.** To generate input features, we use sinusoidal positional embeddings (Vaswani et al., 2017) to embed the time (for agents and dynamic roadgraph) and $xyz$-coordinates separately into a $D$ dimensional features per dimension. We encode the type of each object using a one-hot encoding (e.g. object type, lane type, etc), and concatenate any other features provided in the data set such as yaw, width, length, height, and velocity. Dynamic road graphs have a second one-hot encoding indicating state, like the traffic light state. Lastly, for agents and the dynamic road graph, we add a binary indicator on whether the agent is hidden at that time step. If the agent or dynamic road graph element is hidden, all input features (e.g. position, type, velocity, state, etc) are set to 0 before encoding except the time embedding, which are linearly spaced values at the dataset's update rate starting at 0, and the hidden indicator.

For agents and the dynamic road graph, we use a 2 layer MLP with a hidden and output dimension of $D$ to produce a final feature per agent or object and per time step. For the static road graph, we must reduce a point cloud of up to $20,000$ road graph points, each belonging to a smaller set of polylines, to a single vector per polyline. Because some lanes can be extremely long, we break up any polyline longer than 20 points into a new set of smaller polylines. Then, we apply a small PointNet (Qi et al.,

2017) architecture with a 2 layer MLP with a hidden and output dimension of $D$ to each point, and use max pooling per polyline to get a final feature per element.

**Inference latency.** Scene Transformer produces a prediction for every agent in the scene in a single pass. Its inference speed depends on the number of agents in the scene. Our preliminary profiling of inference speed ranged from 52 ms (32 agents) to 175 ms (128 agents) on the Waymo Open Motion Dataset (128 agents, 91 timesteps for each agent, up to 1400 roadgraph elements). This was measured on an Nvidia V100 using a standard TensorFlow inference graph in float32 without optimization tricks, optimization tuning, etc. This is in line with the expected linear scaling of the factorized attention modules.

| Meta-Arch | Name | Input | Operation | Queries | Keys/Values | Across | Atten Matrix | Output Size | # Param |
|---|---|---|---|---|---|---|---|---|---|
| **Encoder** | | | | | | | | | |
| | $\mathcal{A}$ | Agents | MLP + BN | – | – | – | – | $[A, T, D]$ | 334080 |
| | $\mathcal{B}$ | Dyna RG | MLP + BN | – | – | – | – | $[G_D, T, D]$ | 337408 |
| | $\mathcal{C}$ | Static RG | MLP + BN | – | – | – | – | $[G_S, T, D]$ | 270592 |
| | $\mathcal{D}$ | $\mathcal{A}, \mathcal{B}, \mathcal{C}$ | Transformer | Agents | Agents | Time | $[A, T, T]$ | $[A, T, D]$ | 789824 |
| | $\mathcal{E}$ | $\mathcal{D}$ | Transformer | Agents | Agents | Agents | $[T, A, A]$ | $[A, T, D]$ | 789824 |
| | $\mathcal{F}$ | $\mathcal{E}$ | Transformer | Agents | Agents | Time | $[A, T, T]$ | $[A, T, D]$ | 789824 |
| | $\mathcal{G}$ | $\mathcal{F}$ | Transformer | Agents | Agents | Agents | $[T, A, A]$ | $[A, T, D]$ | 789824 |
| | $\mathcal{H}$ | $\mathcal{G}$ | Transformer | Agents | Agents | Time | $[A, T, T]$ | $[A, T, D]$ | 789824 |
| | $\mathcal{I}$ | $\mathcal{H}$ | Transformer | Agents | Agents | Agents | $[T, A, A]$ | $[A, T, D]$ | 789824 |
| | $\mathcal{J}$ | $\mathcal{I}$ | Transformer | Agents | Static RG | Time | $[T, A, G_S]$ | $[A, T, D]$ | 789824 |
| | $\mathcal{K}$ | $\mathcal{J}$ | Transformer | Agents | Dyna RG | Time | $[T, A, G_D]$ | $[A, T, D]$ | 789824 |
| | $\mathcal{L}$ | $\mathcal{K}$ | Transformer | Agents | Agents | Time | $[A, T, T]$ | $[A, T, D]$ | 789824 |
| | $\mathcal{M}$ | $\mathcal{L}$ | Transformer | Agents | Agents | Agents | $[T, A, A]$ | $[A, T, D]$ | 789824 |
| | $\mathcal{N}$ | $\mathcal{M}$ | Transformer | Agents | Static RG | Time | $[T, A, G_S]$ | $[A, T, D]$ | 789824 |
| | $\mathcal{O}$ | $\mathcal{N}$ | Transformer | Agents | Dyna RG | Time | $[T, A, G_D]$ | $[A, T, D]$ | 789824 |
| | $\mathcal{P}$ | $\mathcal{O}$ | Transformer | Agents | Agents | Time | $[A, T, T]$ | $[A, T, D]$ | 789824 |
| | $\mathcal{Q}$ | $\mathcal{P}$ | Transformer | Agents | Agents | Agents | $[T, A, A]$ | $[A, T, D]$ | 789824 |
| **Decoder** | | | | | | | | | |
| | $\mathcal{R}$ | $\mathcal{Q}$ | Tile | – | – | – | – | $[F, A, T, D]$ | 0 |
| | $\mathcal{S}$ | $\mathcal{R}$ | Concat | – | – | – | – | $[F, A, T, D+F]$ | 0 |
| | $\mathcal{T}$ | $\mathcal{S}$ | MLP | – | – | – | – | $[F, A, T, D]$ | 68096 |
| | $\mathcal{U}$ | $\mathcal{T}$ | Transformer | Agents | Agents | Time | $[A, T, T]$ | $[F, A, T, D]$ | 789824 |
| | $\mathcal{V}$ | $\mathcal{U}$ | Transformer | Agents | Agents | Agents | $[T, A, A]$ | $[F, A, T, D]$ | 789824 |
| | $\mathcal{W}$ | $\mathcal{V}$ | Transformer | Agents | Agents | Time | $[A, T, T]$ | $[F, A, T, D]$ | 789824 |
| | $\mathcal{X}$ | $\mathcal{W}$ | Transformer | Agents | Agents | Agents | $[T, A, A]$ | $[F, A, T, D]$ | 789824 |
| | $\mathcal{Y}$ | $\mathcal{X}$ | Layer Norm | – | – | – | – | $[F, A, T, D]$ | 512 |
| | $\mathcal{Z}_1$ | $\mathcal{Y}$ | MLP + BN | – | – | – | – | $[F, A, T, 6]$ | 66817 |
| | $\mathcal{Z}_2$ | $\mathcal{Y}$ | MLP | – | – | – | – | $[F, A, T, 7]$ | 1799 |

| | |
|---|---|
| Optimizer | Adam ($\alpha = 1e{-}4$, $\beta_1 = 0.9$, $\beta_2 = 0.999$) |
| Learning Rate Schedule | Total epochs: 150; Linear ramp-up: 0.1 epochs |
| Batch size | 64 |
| Gradient Clipping (norm) | 5.0 |
| Weight initialization | Glorot & Bengio (2010) |
| Weight decay | None |
| Position Embeddings | Min Timescale: 4; Max Timescale: 256 |
| Temporal Embeddings | Min Timescale: 6; Max Timescale: 80 |
| Future classification weight | 0.1 |
| Position classification weight | 1.0 |
| Laplace Target Scale | 1.0 |

Table 5: **Scene Transformer architecture and training details.** The network receives as input of $A$ agents across $T$ time steps and $K$ features. $K$ is the total number of input features (e.g. 3-D position, velocity, object type, bounding box size). A subset of these inputs are masked. $G_S$ and $G_D$ is the maximum number of road graph (RG) elements and $D$ is the total number of features. MLP and BN denote multilayer perception and batch normalization (Ioffe & Szegedy, 2015), respectively. The output of the network is $\mathcal{Z}_1$ and $\mathcal{Z}_2$. $\mathcal{Z}_1$ corresponds to predicting the logits for classifying which one of the $F$ futures is most likely. $\mathcal{Z}_2$ corresponds to the predicted $xyz$-coordinates with their associated uncertainties, and a single value for heading. In our model $D = 256$, $K = 7$ and $F = 6$ for a total of 15,296,136 parameters for both datasets. For the Waymo Open Motion Dataset $G_s = 1400$, $G_d = 16$, $T = 91$, $A = 128$, and for Argoverse $G_s = 256$, $G_d = 0$, $T = 50$, and $A = 64$. All layers employ ReLU nonlinearities.

| Name | Input | Operation | Parameter Sizes | Output Size | # Param |
|------|-------|-----------|-----------------|-------------|---------|
| $X$ | $X_o$ | Layer Norm | $[D], [D]$ | $[A, T, D]$ | 512 |
| $K$ | $X$ | Affine Projection | $[D, H, \frac{D}{H}], [H, \frac{D}{H}]$ | $[A, T, H, \frac{D}{H}]$ | 65792 |
| $V$ | $X$ | Affine Projection | $[D, H, \frac{D}{H}], [H, \frac{D}{H}]$ | $[A, T, H, \frac{D}{H}]$ | 65792 |
| $Q_o$ | $X$ | Affine Projection | $[D, H, \frac{D}{H}], [H, \frac{D}{H}]$ | $[A, T, H, \frac{D}{H}]$ | 65792 |
| $Q$ | $Q_o$ | Rescale | $[\frac{D}{H}]$ | $[A, T, H, \frac{D}{H}]$ | 64 |
| $Y_1$ | $Q, K, V$ | $\mathsf{softmax}(Q\,K^T)V$ | $-$ | $[A, T, H, \frac{D}{H}]$ | 0 |
| $Y_2$ | $Y_1$ | Affine Projection | $[H, \frac{D}{H}, D], [D]$ | $[A, T, D]$ | 65792 |
| $S$ | $F_1, X_o$ | Sum | $-$ | $[A, T, D]$ | 0 |
| $F_1$ | $Y_2$ | MLP | $[D, kD], [kD]$ | $[A, T, kD]$ | 263168 |
| $F_2$ | $S$ | MLP | $[kD, D], [D]$ | $[A, T, D]$ | 262400 |
| $Z$ | $F_2$ | Layer Norm | $[D], [D]$ | $[A, T, D]$ | 512 |
| **Total** | | | | | 789824 |

Table 6: **Transformer architecture.** The network receives as input $X_o$ and outputs $Z$. All MLP's employ a ReLU nonlinearity. $D$ is the number of feature dimensions; $H$ is the number of attention heads. In our model $D$=256, $H$=4 and $k$=4.

# B  ADDITIONAL MOTION PREDICTION RESULTS

| Motion Prediction | minADE↓ | | | minFDE↓ | | | MR↓ | | | mAP↑ | | |
|---|---|---|---|---|---|---|---|---|---|---|---|---|
| | veh | ped | cyc | veh | ped | cyc | veh | ped | cyc | veh | ped | cyc |
| *valid* | | | | | | | | | | | | |
| baseline (Ettinger et al., 2021) | 0.39 | 0.19 | 0.41 | 0.65 | 0.36 | 0.73 | 0.14 | 0.07 | 0.25 | 0.33 | **0.33** | 0.27 |
| ours (marginal) | **0.33** | 0.20 | **0.39** | **0.57** | **0.33** | **0.67** | **0.11** | **0.07** | **0.21** | **0.38** | 0.33 | **0.28** |
| ours (joint-as-marginal) | 0.42 | 0.28 | 0.50 | 0.78 | 0.51 | 0.94 | 0.19 | 0.21 | 0.32 | 0.34 | 0.25 | 0.23 |
| ours (multi-task joint-as-marginal) | 0.43 | 0.51 | 0.51 | 0.80 | 0.52 | 0.93 | 0.20 | 0.21 | 0.32 | 0.33 | 0.24 | 0.23 |
| *test* | | | | | | | | | | | | |
| baseline (Ettinger et al., 2021) | 0.39 | 0.20 | 0.41 | 0.65 | 0.36 | 0.74 | 0.14 | 0.07 | 0.25 | 0.34 | **0.32** | 0.24 |
| ours (marginal) | **0.32** | 0.20 | **0.38** | **0.56** | **0.33** | **0.67** | **0.11** | **0.07** | **0.21** | **0.38** | 0.32 | **0.28** |
| ours (joint-as-marginal) | 0.42 | 0.28 | 0.49 | 0.78 | 0.53 | 0.92 | 0.19 | 0.21 | 0.32 | 0.33 | 0.26 | 0.24 |
| ours (multi-task joint-as-marginal) | 0.44 | 0.29 | 0.50 | 0.80 | 0.53 | 0.93 | 0.20 | 0.22 | 0.32 | 0.32 | 0.24 | 0.24 |

Table 7: **Marginal predictive performance on Waymo Open Motion Dataset motion prediction for** $t = 3$ **seconds**. Please see Table 2 for details.

| Interaction Prediction | minSADE↓ | | | minSFDE↓ | | | SMR↓ | | | mAP↑ | | |
|---|---|---|---|---|---|---|---|---|---|---|---|---|
| | veh | ped | cyc | veh | ped | cyc | veh | ped | cyc | veh | ped | cyc |
| *valid* | | | | | | | | | | | | |
| baseline (Ettinger et al., 2021) | 0.58 | 0.43 | 0.60 | 1.13 | 0.86 | 1.20 | 0.45 | 0.47 | 0.61 | 0.15 | 0.13 | 0.06 |
| ours (marginal-as-joint) | 0.45 | 0.37 | 0.52 | 0.91 | 0.75 | 1.04 | 0.35 | 0.42 | 0.52 | 0.20 | 0.12 | 0.09 |
| ours (joint) | 0.41 | **0.34** | **0.47** | 0.81 | **0.65** | 0.92 | **0.29** | **0.38** | **0.49** | 0.26 | 0.14 | 0.11 |
| ours (multi-task joint) | **0.40** | 0.34 | **0.47** | **0.80** | **0.65** | **0.91** | **0.28** | **0.38** | **0.49** | **0.27** | **0.17** | **0.13** |
| *test* | | | | | | | | | | | | |
| baseline (Ettinger et al., 2021) | 0.58 | 0.42 | 0.61 | 1.14 | 0.85 | 1.21 | 0.45 | 0.47 | 0.61 | 0.16 | 0.11 | 0.05 |
| ours (marginal-as-joint) | 0.45 | 0.36 | 0.53 | 0.93 | 0.74 | 1.06 | 0.36 | 0.40 | 0.55 | 0.18 | 0.11 | 0.07 |
| ours (joint) | **0.41** | **0.33** | **0.48** | 0.82 | **0.64** | **0.94** | **0.29** | **0.36** | **0.50** | 0.18 | 0.12 | 0.07 |
| ours (multi-task joint) | **0.41** | 0.34 | **0.48** | **0.81** | 0.65 | **0.94** | **0.29** | 0.37 | 0.51 | **0.26** | **0.15** | **0.10** |

Table 8: **Joint predictive performance on Waymo Open Motion Dataset motion prediction for** $t = 3$ **seconds**. Please see Table 3 for details.

| Motion Prediction | minADE↓ | | | minFDE↓ | | | MR↓ | | | mAP↑ | | |
|---|---|---|---|---|---|---|---|---|---|---|---|---|
| | veh | ped | cyc | veh | ped | cyc | veh | ped | cyc | veh | ped | cyc |
| *valid* | | | | | | | | | | | | |
| baseline (Ettinger et al., 2021) | 0.74 | 0.37 | 0.75 | 1.36 | 0.73 | 1.43 | 0.17 | 0.10 | 0.25 | 0.29 | **0.27** | **0.26** |
| ours (marginal) | **0.65** | **0.35** | **0.69** | **1.23** | **0.67** | **1.30** | **0.15** | **0.10** | **0.22** | **0.33** | 0.26 | 0.25 |
| ours (joint-as-marginal) | 0.83 | 0.51 | 0.93 | 1.68 | 1.08 | 1.91 | 0.23 | 0.26 | 0.33 | 0.26 | 0.19 | 0.18 |
| ours (multi-task joint-as-marginal) | 0.84 | 0.52 | 0.92 | 1.72 | 1.08 | 1.87 | 0.24 | 0.26 | 0.33 | 0.26 | 0.18 | 0.18 |
| *test* | | | | | | | | | | | | |
| baseline (Ettinger et al., 2021) | 0.74 | 0.37 | 0.76 | 1.35 | 0.73 | 1.43 | 0.17 | 0.10 | 0.25 | 0.29 | **0.26** | **0.23** |
| ours (marginal) | **0.63** | **0.35** | **0.68** | **1.20** | **0.67** | 1.31 | **0.11** | **0.10** | **0.22** | **0.33** | 0.26 | **0.23** |
| ours (joint-as-marginal) | 0.83 | 0.52 | 0.91 | 1.68 | 1.10 | 1.88 | 0.23 | 0.26 | 0.33 | 0.26 | 0.19 | 0.22 |
| ours (multi-task joint-as-marginal) | 0.85 | 0.53 | 0.93 | 1.72 | 1.10 | 1.91 | 0.24 | 0.27 | 0.34 | 0.26 | 0.19 | 0.21 |

Table 9: **Marginal predictive performance on Waymo Open Motion Dataset motion prediction for** $t = 5$ **seconds**. Please see Table 2 for details.

| Interaction Prediction | minSADE↓ | | | minSFDE↓ | | | SMR↓ | | | mAP↑ | | |
|---|---|---|---|---|---|---|---|---|---|---|---|---|
| | veh | ped | cyc | veh | ped | cyc | veh | ped | cyc | veh | ped | cyc |
| *valid* | | | | | | | | | | | | |
| baseline (Ettinger et al., 2021) | 1.19 | 0.90 | 1.25 | 2.64 | 1.98 | 2.82 | 0.55 | 0.57 | 0.70 | 0.13 | 0.09 | 0.04 |
| ours (marginal-as-joint) | 0.96 | 0.78 | 1.08 | 2.15 | 1.75 | 2.46 | 0.44 | 0.51 | 0.60 | 0.13 | 0.09 | 0.06 |
| ours (joint) | 0.84 | 0.68 | 0.95 | 1.81 | 1.45 | 2.09 | **0.38** | **0.48** | **0.59** | 0.18 | 0.08 | **0.07** |
| ours (multi-task joint) | **0.83** | **0.68** | **0.95** | 1.79 | 1.45 | **2.08** | **0.38** | 0.48 | **0.59** | **0.19** | **0.10** | 0.06 |
| *test* | | | | | | | | | | | | |
| baseline (Ettinger et al., 2021) | 1.21 | 0.89 | 1.26 | 2.70 | 1.96 | 2.80 | 0.56 | 0.59 | 0.69 | 0.13 | 0.07 | 0.03 |
| ours (marginal-as-joint) | 0.96 | 0.77 | 1.07 | 2.19 | 1.74 | 2.45 | 0.46 | 0.51 | 0.62 | 0.13 | 0.09 | 0.06 |
| ours (joint) | 0.85 | **0.67** | **0.96** | 1.85 | **1.44** | 2.10 | **0.39** | 0.50 | **0.58** | 0.17 | **0.10** | 0.07 |
| ours (multi-task joint) | **0.84** | 0.68 | 0.97 | **1.82** | 1.46 | **2.09** | **0.39** | 0.49 | 0.59 | **0.20** | 0.09 | **0.08** |

Table 10: **Joint predictive performance on Waymo Open Motion Dataset motion prediction for** $t = 5$ **seconds**. Please see Table 3 for details.

| Motion Prediction | minADE↓ | | | minFDE↓ | | | MR↓ | | | mAP↑ | | |
|---|---|---|---|---|---|---|---|---|---|---|---|---|
| | veh | ped | cyc | veh | ped | cyc | veh | ped | cyc | veh | ped | cyc |
| *valid* | | | | | | | | | | | | |
| LSTM baseline (Ettinger et al., 2021) | 1.34 | 0.63 | 1.26 | 2.85 | 1.35 | 2.68 | 0.25 | 0.13 | 0.29 | 0.23 | 0.23 | 0.20 |
| ours (marginal) | **1.17** | **0.59** | **1.15** | **2.51** | **1.26** | **2.44** | **0.20** | **0.12** | **0.24** | 0.26 | 0.27 | 0.20 |
| ours (joint-as-marginal) | 1.53 | 0.90 | 1.63 | 3.48 | 2.09 | 3.77 | 0.28 | 0.30 | 0.37 | 0.20 | 0.16 | 0.13 |
| ours (multi-task joint-as-marginal) | 1.56 | 0.89 | 1.60 | 3.56 | 2.06 | 3.68 | 0.29 | 0.29 | 0.37 | 0.19 | 0.15 | 0.15 |
| *test* | | | | | | | | | | | | |
| LSTM baseline (Ettinger et al., 2021) | 1.34 | 0.64 | 1.29 | 2.83 | 1.35 | 2.68 | 0.24 | 0.13 | 0.29 | 0.24 | 0.22 | 0.19 |
| ours (marginal) | **1.17** | **0.60** | **1.17** | **2.48** | **1.25** | **2.43** | **0.19** | **0.12** | **0.22** | 0.27 | 0.23 | 0.20 |
| ours (joint-as-marginal) | 1.52 | 0.91 | 1.61 | 3.43 | 2.09 | 3.68 | 0.28 | 0.30 | 0.37 | 0.19 | 0.16 | 0.19 |
| ours (multi-task joint-as-marginal) | 1.55 | 0.91 | 1.64 | 3.50 | 2.08 | 3.75 | 0.28 | 0.29 | 0.38 | 0.19 | 0.16 | 0.19 |

Table 11: **Marginal predictive performance on Waymo Open Motion Dataset motion prediction for** $t = 8$ **seconds**. Please see Table 2 for details. We additionally include "joint-as-marginal" results for our standard MP masked joint model, and our multi-task joint model. These are joint models evaluated as if they were marginal models (with no changes in the outputs).

| Interaction Prediction | minSADE↓ | | | minSFDE↓ | | | SMR↓ | | | mAP↑ | | |
|---|---|---|---|---|---|---|---|---|---|---|---|---|
| | veh | ped | cyc | veh | ped | cyc | veh | ped | cyc | veh | ped | cyc |
| *valid* | | | | | | | | | | | | |
| LSTM baseline (Ettinger et al., 2021) | 2.42 | 2.73 | 3.16 | 6.07 | 4.20 | 6.46 | 0.66 | 1.00 | 0.83 | 0.07 | **0.06** | 0.02 |
| ours (marginal-as-joint) | 2.04 | 1.62 | 2.28 | 4.94 | 3.81 | 5.67 | 0.54 | 0.63 | 0.72 | **0.11** | 0.05 | 0.03 |
| ours (joint, mp-only) | **1.72** | **1.38** | 1.96 | **3.98** | **3.11** | 4.75 | **0.49** | **0.60** | 0.73 | **0.11** | 0.05 | 0.03 |
| ours (joint, multi-task) | **1.72** | 1.39 | **1.94** | 3.99 | 3.15 | **4.69** | **0.49** | 0.62 | **0.71** | **0.11** | **0.06** | **0.04** |
| *test* | | | | | | | | | | | | |
| LSTM baseline (Ettinger et al., 2021) | 2.46 | 2.47 | 2.96 | 6.22 | 4.30 | 6.26 | 0.67 | 0.89 | 0.89 | 0.06 | 0.03 | 0.03 |
| ours (marginal-as-joint) | 2.08 | 1.62 | 2.24 | 5.04 | 3.87 | 5.41 | 0.55 | 0.64 | 0.73 | 0.08 | **0.05** | 0.03 |
| ours (joint, mp-only) | 1.76 | **1.38** | 1.95 | 4.08 | **3.19** | 4.65 | **0.50** | 0.62 | **0.70** | 0.10 | **0.05** | **0.04** |
| ours (joint, multi-task) | **1.74** | 1.41 | **1.95** | **4.06** | 3.26 | 4.68 | **0.50** | 0.64 | 0.71 | **0.13** | 0.04 | 0.03 |

Table 12: **Joint predictive performance on Waymo Open Motion Dataset motion prediction for** $t = 8$ **seconds**. These results are the same as Table 3 but is included here for completeness with the other Appendix tables.

## B.1 UNDERSTANDING THE TRADE-OFFS BETWEEN THE JOINT AND MARGINAL MODELS

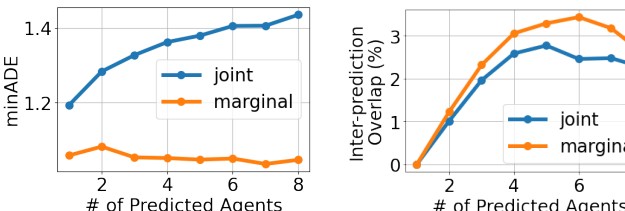

Figure 5: **Quantitative comparison of the marginal and joint prediction models.** Left: The *marginal* minADE broken down as a function of number of predicted agents. When there are more agents, producing internally consistent predictions is more challenging, and hence the joint model performs slightly worse. Right: Overlap rate between pairs of agent predictions. The joint model produces internally consistent predictions with lower inter-prediction overlap rates.

While the joint mp-only model performs better on the interactive prediction task than the marginal model *when evaluated with joint metrics*, it performs worse on the motion prediction task *when evaluated with marginal metrics* (see Table 11 "ours (joint)"). This is because the joint model is trained to produce predictions that are internally consistent between agents, while the marginal model does not, which is a strictly harder task. In this section, we examine this quality difference and the internal consistency of the predictions from both models.

In the WOMD dataset, each example has a different number of agents to be predicted. By slicing the *marginal* minADE results based on the number of agents to be predicted (Figure 5, left), we find that the joint model performs worse as there are more predicted agents, while the marginal model performs the same. This is expected: when there are more agents, the joint model has a more difficult task since it needs to produce internally consistent predictions. One may expect a joint model to need an exponential number of trajectory combinations to perform competitively with the marginal model, but we are encouraged to find that this is not actually the case. We believe this is due to the fact that when many agents are interacting, the number of realistic scenarios is actually heavily constrained by interactions - most combinations of marginal predictions don't actually make sense together.

However, when more agents are to be predicted, the possibility of interactions are higher and we would expect that the joint model is able to capture these interactions through internally consistent predictions. We measure the models' internal consistency by selecting the best trajectory and measuring the *inter-prediction overlap rate* (details below). We find that the joint model has a consistently lower inter-prediction overlap rate, showing that it is able to capture the interactions between agents. The ability to model interactions enables the model to be suitable for conditional motion prediction, and goal conditioned prediction tasks, which we discuss in Section 4.4.

**Measuring agent inter-prediction overlap.** We measure *predicted* agent overlap on the best set of trajectory produced by the model. For the joint model, this corresponds to the joint prediction that has the higher probability score. For the marginal model, we take the top scoring trajectory for each agent to obtain the best set. For every predicted agent in the trajectory, we determine if it has an overlap with any *other predicted* agent by comparing the rotated upright 3D bounding boxes using our predicted xyz and heading. The inter-prediction overlap rate is the number of predicted agents that are involved in some overlap, divided by the total number of predicted agents. We count two agents as overlapping if the intersection over the union (IoU) exceeds 0.01.

# C   SLICING RESULTS

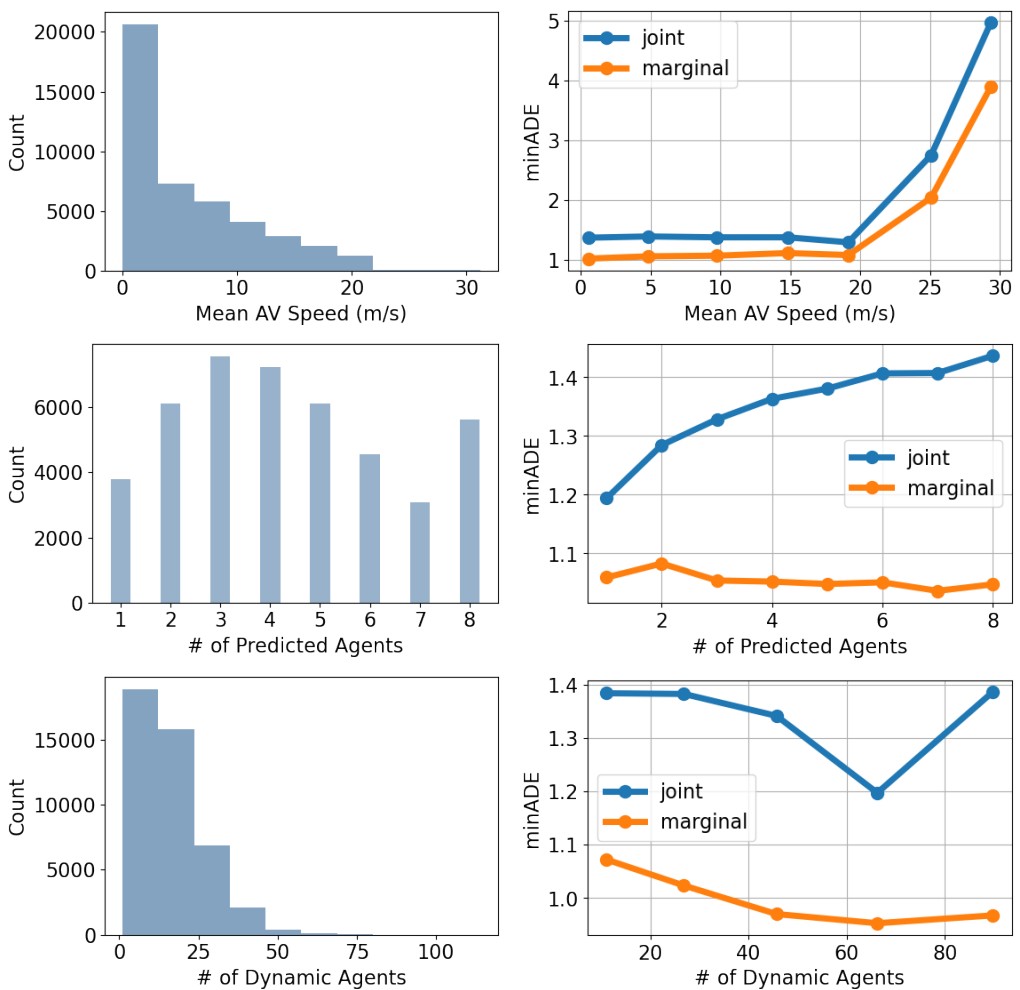

Figure 6: We show marginal minADE on a per scene basis broken down by different scene level statistics. Interestingly, we show that both the marginal and joint models become out of distribution above 20 m/s, where there is minimal training data.

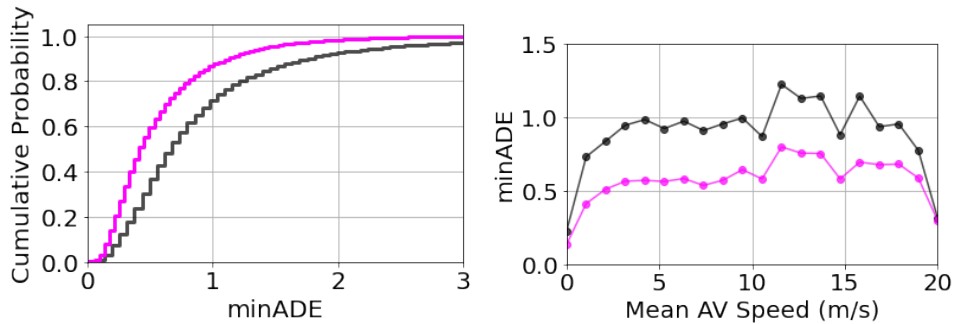

Figure 7: **Analysis of the multi-task model for goal-conditioned motion prediction.** Left: Cumulative probability of the AV minADE in goal conditioned prediction (blue) and motion prediction (black) masking strategies. Right: AV minADE for goal conditioned prediction (blue) and motion prediction (black) as a function of AV speed (averaged over the ground truth trajectory).

# D  LOSS IMPLEMENTATION

The Scene Transformer model is both agent permutation equivariant and scene-centric. These properties allow us to easily switch between a marginal vs joint loss formulation (Figure 8). In the marginal formulation, we reduce the loss of the best future for every agent separately; while in the joint formulation, we reduce the loss of the best future for *all* agents jointly. In practice, this determines when the reduce_min operation is performed.

Our final loss composes the regression loss on the trajectory and the classification loss of the best trajectory. A weighted linear combination of the loss terms is used to combine these two losses together. The classification loss weight was set to be 0.1, while the regression losses have weight 1.0. The weights were determined using the hold-out validation set.

```python
# agent_predictions and agent_gt are
# [F, A, T, 7] Tensors with [x, y, z], uncertainty terms, and yaw.

# Get the KL Divergence of the predictions vs ground truth.
# Use LaplaceKL method from (Meyer & Thakurdesai, 2020)
loss = LaplaceKL(agent_predictions, agent_gt)

# Now reduce across all timesteps and values to produce
# a tensor of shape [F, A]
loss = tf.reduce_sum(loss, axis=[2, 3])

# The marginal loss, we only apply the loss to the best trajectory
# per agent (so min across the future dimension).
marginal_loss = tf.reduce_min(loss, axis=0)

# Then sum over the agent dimension
marginal_loss = tf.reduce_sum(marginal_loss)

# The joint loss, we sum over all agents to get a loss value
# per future.
joint_loss = tf.reduce_sum(loss, axis=1)

# Then only apply the loss to the best future prediction
joint_loss = tf.reduce_min(joint_loss)
```

Figure 8: Pseudo-code in TensorFlow (Abadi et al., 2015) demonstrating the joint versus marginal loss formulation.

