# OpenReview forum: "Scene Transformer: A unified architecture for predicting future trajectories of multiple agents"
_ICLR.cc/2022/Conference — ICLR 2022 Poster_

### Official Review · Reviewer_RDXN · 2021-10-18

**Correctness:** 3
**Technical Novelty And Significance:** 3
**Empirical Novelty And Significance:** 3
**Recommendation:** 8
**Confidence:** 5

**Main Review:**

--- Strengths

- This is a very thought-provoking paper with many interesting novel ideas. It uses a transformer architecture to translate the input [A, T, D] tensor into the prediction output [F, A, T, 7] tensor. With this design, it can perform motion prediction, conditioned motion prediction, and goal-conditioned motion prediction all in the same model. It also proposes a factorized self-attention approach to reduce the computational cost.

- The result is quite impressive. It achieves state-of-the-art performance on both Argoverse and Waymo Open Motion Dataset (WOMD).


--- Issues and suggestions

- This paper proposes a method to model the joint distribution of the trajectories of all agents in the scene, and this is listed as one of its three contributions. However, I don't think this is a significant contribution of this paper because 1) the idea is fairly straightforward and 2) the performance is questionable. The difference between training a marginal model vs. a joint model is the loss function. The loss of a marginal model is sum_A(min_F(sum_T(error))), while the loss of a joint model is min_F(sum_A(sum_T(error))). This loss function design is actually fairly straightforward, and every model that predicts trajectories of all agents in the scene can do that. What makes joint trajectory prediction hard is not this loss function design but the challenge of scaling with the number of agents in the scene. A crowded scene can easily have up to 100 agents, and simply modeling the joint prediction this way won't work in practice because it requires F to be very large. I expected this paper to tackle this challenge, but I don't see it in the paper. The WOMD Interaction Prediction task only requires the model to predict the joint prediction of two agents, so this simple approach works there, but I just don't see how this approach can work in an actual autonomous driving software stack. Also, in Figure 4 in the appendix, the prediction error of the joint model looks terrible even with a few (e.g., four) agents.

- The appendix mentions that the model was trained with some data augmentation. So it's not very clear to me how much of the performance improvement comes from the Scene Transformer design and how much of it comes from data augmentation. It will be useful to also show the performance numbers without data augmentation.

- It would be helpful to show some inference latency numbers for people to understand whether the proposed model can be used in practice. Also, this paper mentions using the factorized self-attention to save computation, so it will be helpful to show some numbers on that as well.

- A few statements in this paper are pretty confusing and hard to understand. I believe there is some room for writing improvements there.  I will give a few examples below.

- I don't understand why "two agents of the same type hidden at the same future location will have identical input representations". Don't they already have different past positions unless they are on top of each other?

- The paper argues that it uses a scene-centric representation while the prediction space is agent-centric, but I find a lot of statements around this to be quite confusing. For example, in Section 2, it says, "Some models do a majority of modeling in a global, scene-level coordinate frame, such as work that employs a rasterized top-down image". However, I believe most of the rasterization-based prediction methods are actually agent-centric. They center the raster around the agent-of-interest (e.g., in Cui et al., 2019). Actually, there are a lot more works than just LaneGCN that predict trajectories in agent-centric frame from scene-centric features, such as MultiXNet (Djuric el al. 2020) and SPAGNN (Casas el al. 2020). In Section 3.1, the sentence "We use a scene-centric embedding where we use an agent of interest’s position as the origin, and encode all roadgraph and agents with respect to it." is quite confusing. People will wonder why it is not agent-centric when you use the agent of interest as the origin. I think what the authors are trying to say is that they use the ego vehicle as the origin in Waymo Open Motion Dataset (WOMD), so it's scene-centric. But I think for Argovese dataset, it is still agent-centric. On the WOMD dataset, where the ego vehicle is the origin, do you predict trajectory waypoints of an agent with respect to the ego vehicle or the agent? I think the trajectory waypoints should also be with respect to the ego vehicle if you want to model agent-agent interactions, but this seems to contradict the statement of "agent-centric representations". This part is very unclear in the paper, and please clarify this.

- The paper says the 7 output values of each waypoint are x/y position and uncertainty in 3D space and heading. But if we want to model the output waypoints in 3D space, shouldn't the heading be 3D yaw/pitch/roll heading? The appendix says the losses and uncertainties are defined in lateral and longitudinal coordinates, and this will only make sense if it's a 2D space. This really makes me wonder whether the output space is 3D or 2D. Also, I don't think it's a good idea to represent the heading with just one value, because, for example, when the model predicts a 179 deg heading while the ground-truth heading is -179 deg, it will get a very high loss. I doubt how this can work.

- In the appendix, the paper says, "We append on a one-hot encoding to the final dimension where a 1 indicates which of the F futures is the ground truth trajectory resulting in a tensor of shape [F, A, T, D + F]." This can only be done at training time, and I don't get how we can get the [F, A, T, D + F] tensor at inference time.


**Summary Of The Paper:**

This paper proposes a transformer-style architecture, Scene Transformer, for joint trajectory prediction of multiple agents (including the autonomous vehicle). The key representation of Scene Transformer is a 3D tensor [A, T, D], with the dimensions being agent, time step, and feature dimension, and we mask out the entries that to be predicted. The output is [F, A, T, 7], where F is the number of future predictions. The model outputs 7 values for each waypoint, x/y position and uncertainty in 3D space and heading. The model uses factorized self-attention applied on the A axis and T axis separately to transform the [A, T, D] representation tensor, and uses a cross attention to fuse in the static map features. The model can be trained to predict either marginal predictions or joint predictions depending on the loss function. Also, based on what entries in the input tensor are masked out, Scene Transformer can perform motion prediction, conditioned motion prediction, and goal-conditioned motion prediction all in the same model.

The Scene Transformer model was evaluated on Argoverse and Waymo Open Motion Dataset (WOMD), and its marginal prediction achieves the state-of-the-art performance on both datasets. The evaluation also shows that the model is able to perform joint prediction for the WOMD Interaction Prediction task and achieves better performance on the scene-level metrics compared to the baselines. The evaluation also shows that the Scene Transformer model is able to do conditional motion prediction and goal-conditioned motion prediction.

**Summary Of The Review:**

Overall, I think this is a very thought-provoking paper with many interesting novel ideas and impressive results. The performance of the marginal prediction is a bit questionable, and there is also room for some writing improvement. I recommend this paper to be accepted, but I hope the authors can clarify my questions in the final version of the paper.

---

> ### Author Response · Authors · 2021-11-16
> **Response to Reviewer RDXN**
>
> We want to thank the reviewer for their extensive and detailed review, and thoughtful questions/comments. In particular, we appreciate the reviewer noting that “This is a very thought-provoking paper with many interesting novel ideas.”, and recognizing our strong results.
>
> We provide some responses to the issues and suggestions below:
>
> **Loss formulation**: We agree with the reviewer that the change in the loss function formulation from marginal to joint is straightforward in hindsight. One of the reasons why this works so well is the careful design of our model to have all agents and roadgraph in a shared, scene-centric coordinate system. The fact that a simple formulation works well enough to obtain SOTA on the Waymo interactive challenge makes this more valuable. It turns out that it’s not obvious that this simple formulation will work for all models; specifically, traditional agent-centric trajectory prediction models may not so easily be turned into joint models via this simple approach.
>
> **On the number of joint predictions**: The reviewer makes a good point. While it might be intuitive that a large number of futures may be needed, we don’t think it may be necessary. If agents are interacting, those interactions can constrain the set of possible trajectories that are valid for a joint model, making the realistic set of F to be very small. We show an initial promising indication of this in the Appendix (see Figure 4): as the number of agents we have to predict with the joint model increases, the minADE also increases, but it does so in a manner that looks more linear or sub-linear.
>
> **Data augmentation**: We found data augmentation to be important, but not crucial, to obtain good results. We primarily use it to combat overfitting and regularize the models. Between the review being posted and this rebuttal, we ran a simple experiment to compare the marginal Waymo Open Motion Dataset model with our two data augmentations, versus without any data augmentation. The results are on the normal motion prediction validation set. It’s worth noting that it’s hard to make any conclusions here, because we don’t have results for other prior work with and without these augmentations. The results are shared below, and we will happily add a short appendix section discussing these results. We note that our validation and test set numbers almost exactly match, and if that holds, a Scene Transformer without data augmentation would still be SOTA among published methods.
>
> | With Augmentation?      | Vehicle minADE @ 8s | Pedestrian minADE @ 8s | Cyclist minADE @ 8s |
> | ----------- | ----------- | ----------- | ----------- |
> | No      | 1.298       | 0.687 | 1.291 |
> | Yes   | 1.178        | 0.592 | 1.153 |
>
>
> **Inference latency**: SceneTransformer produces a prediction for every agent in the scene in a single pass. Its inference speed depends on the number of agents in the scene. Our preliminary profiling of inference speed ranged from 52 ms (32 agents) to 175 ms (128 agents) on the Waymo Open Motion Dataset (128 agents, 91 timesteps for each agent, up to 1400 roadgraph elements). This was measured on a Nvidia V100 using a standard Tensorflow inference graph in float32 without optimization tricks, optimization tuning, etc. This is in line with the expected linear scaling of the factorized attention modules. We are happy to have a short appendix section with these results.
>
> **On identical input representations**: This is a problem if we were to do full self attention instead of factorized self attention. If we were to do full attention, every agent-timestep feature (e.g. a specific vector of size F in [A, T, F]) that is masked and of the same type will have the exact same features, which led to training instability during our early exploration. This stems from the fact that we do not have an agent identity indicator in the feature representation. Using factorized self attention ensures the first attention operation is across timesteps for each agent, allowing each agent's past locations to be encoded into all the other features for that agent (including masked ones). We will work to improve the clarity here in the paper.
>
> **On confusion between scene-centric and agent-centric predictions**: We’ll work to improve the language here in the paper, we agree it can be a bit confusing without simply seeing the value ranges (e.g. all inputs and outputs range from -100 meters to 100 meters in X and Y). Our models inputs and outputs are scene-centric, as in they are absolute meters from the agent at the origin. You are correct that in the Waymo Open Motion Dataset, this is the ego vehicle, and in Argoverse this is just the agent the task asks us to predict (which makes it agent-centric of sorts, but we train on other agents that are in the same scene’s coordinate system). Our outputs are all with respect to the ego vehicle in the scene, so for far away agents that may be values that are in the hundreds of meters.

---

> > ### Author Response · Authors · 2021-11-16
> > **Response to Reviewer RDXN (continued)**
> >
> >
> > **On 7D vs. 9D outputs**: Neither dataset provides pitch/roll values to supervise on. This is similar to a lot of 3D detection datasets that use a 7 degree of freedom representation (XYZ center location, HWL box dimensions, and heading). So in a sense, the output space is somewhere between 2D and fully 3D. Whether the community should move towards a full 3D representation is an interesting open question.
> >
> > **On heading representation as one value**: We’ll add a brief section on this in the Appendix. It turns out that there are variants popular in the 3D detection community that work fine. What we found effective is to wrap the heading error value between [-pi, pi] before calculating our loss. So our loss looks like:
> >
> > ```
> > heading_error = WrapAngleRadians(pred_heading - gt_heading, min_val=-PI, max_val=PI)
> > heading_loss = HuberLoss(heading_error)
> > ```
> >
> > **On one-hot encoding for F futures**: We’ll work to clarify this in the paper. The 1 in the final dimension does not indicate which future is the ground truth. The model does not get any information about the GT trajectory at training or inference time; we simply decode F futures to provide F different joint (or marginal) outputs of the model, and use a one-hot indicator during decoding to make each future’s decoder features slightly different. The ground truth future is matched to the predictions during training by finding which prediction has the lowest loss.

---

> > > ### Comment · Reviewer_RDXN · 2021-11-29
> > > **Thank you for your response**
> > >
> > > Thank you for your response. It clarifies my questions.

---

### Official Review · Reviewer_evQV · 2021-11-02

**Correctness:** 4
**Technical Novelty And Significance:** 2
**Empirical Novelty And Significance:** 2
**Recommendation:** 6
**Confidence:** 5

**Main Review:**

**Strengths:**
1. Paper is well written and clear.
2. The problem statement is justified and relevant in autonomous driving.
3. Results are good.
4. I like the flexibility that the model offers in terms of use cases.

**Weaknesses:**

While the problem statement is important, the technical novelty and the analysis is lacking. The autonomous driving community has started to move beyond rudimentary metrics like ADE/FDE to include more analysis on “interesting behaviors”. For instance, see [0]. Waymo, in fact, even has included such labels in their Open Motion dataset for more challenging scenarios (See Section 3.1 in https://arxiv.org/pdf/2104.10133.pdf).

From a research contribution perspective, it is not enough to simply rack up higher numbers on the leaderboard. There needs to be analysis that advances the field forward. The future of the field lies in challenging scenarios like intersections, unprotected left turns etc. The proposed work currently does not present any such analysis. Therefore, in its present form, the current work in my opinion is not of high technical quality and novelty enough to be published in ICLR.

**What I would like to see in a revised version:**

To improve its novelty this paper needs to include a section in Section 4 on analysis of the proposed method on more challenging and interesting scenarios in the WOMD. Such a section should include:

1. ADE/FDE results for the scenarios that WOMD labels as challenging or interesting.
2. Explanations for where the model particularly is successful and also the cases where the model fails (if so).
3. Furthermore, there is past research on this [1,2,3,4]. The authors should compare and contrast the technical details with these methods in this section.

[0] Ivanovic, Boris, and Marco Pavone. "Rethinking trajectory forecasting evaluation." arXiv preprint arXiv:2107.10297 (2021).

[1]  Deo, Nachiket, and Mohan M. Trivedi. "Trajectory forecasts in unknown environments conditioned on grid-based plans." arXiv preprint arXiv:2001.00735 (2020).

[2] Bhattacharyya, Apratim, Mario Fritz, and Bernt Schiele. "Long-term on-board prediction of people in traffic scenes under uncertainty." Proceedings of the IEEE Conference on Computer Vision and Pattern Recognition. 2018.

[3] Chandra, Rohan, et al. "Traphic: Trajectory prediction in dense and heterogeneous traffic using weighted interactions." Proceedings of the IEEE/CVF Conference on Computer Vision and Pattern Recognition. 2019.

[4] Chandra, Rohan, et al. "Forecasting trajectory and behavior of road-agents using spectral clustering in graph-lstms." IEEE Robotics and Automation Letters 5.3 (2020): 4882-4890.


**Summary Of The Paper:**

The paper proposes a new Transformer-based trajectory forecasting model that can predict multiple agents in a scene. It can be used for goal-directed trajectory forecasting as well. Experiments on Argoverse and Waymo with good results. Related work is sufficient. Technical novelty is low. Paper is clearly written.

**Summary Of The Review:**

Without the required analysis as requested in the revised version, the present work lacks research rigor, technical novelty, and is simply an impressive engineering effort. If no revision is made, I would defer it to a workshop publication.

If authors provide **strong** analysis on challenging, rare, and interesting examples as provided in WOMD, I may recommend acceptance (pending additional review of the provided analysis).

============= POST AUTHOR-DISCUSSION =============================

The authors have made good faith attempt in addressing my concern and engaging in discussion.

While I wouldn't fight for it's acceptance, I have no problems if it is published in ICLR. I have updated my score accordingly.

---

> ### Author Response · Authors · 2021-11-16
> **Response to Reviewer evQV**
>
> Thank you for the review and the suggestions for improvement.  While we agree with the reviewer that the evaluation metrics for motion prediction would benefit from a rethinking [0], we highlight that the goal of this work was to provide a novel perspective on the problem formulation and a connection to recent natural language models.   We further address the comments about the lack of novelty and the lack of analysis.
>
> *On lack of novelty*: the reviewer states “To improve its novelty this paper needs to include a section in Section 4 on analysis…”.  Our paper’s contributions are not simply that we have good numbers; we would like to emphasize the main technical contributions around the problem formulation of driving using masked-language modeling approaches to producing *joint* future predictions.  To our knowledge, no one has assembled the various components of our model in such a simple, parsimonious way; that it provides SOTA performance on the joint task is simply an empirical validation of the choices.
>
> *On analysis*: We completely agree with the reviewer that the autonomous driving community should move beyond aggregate metrics and analyze model performance across rare or interesting slices of data.
>
> However, the Waymo Open Motion Dataset does not actually have labels, metadata, or slices on difficult or interesting scenarios like the reviewer requests. Geographic information is also not provided, nor are labels of intersections or other scenarios of interest, so many interesting types of slicing are simply impossible. We would like to point out that our proposed work does present this type of analysis to the extent possible on the existing dataset.
>
> What the Waymo Open Motion Dataset does have is an Interactive Validation Set that was mined for interesting examples where *exactly 2* agents are interacting with each other. We evaluate all our models on both the original and the interactive versions of the validation dataset and these results are presented in the paper.
>
> Despite the lack of metadata on WOMD, we did make an effort to compute interesting statistics about each example from the data we had available (such as number of moving agents, number of agents model has to predict, SDC speed) and we compared the joint and marginal model performance across these slices:
>
> 1) In Appendix B.1, we analyze the tradeoffs between the joint and marginal model as we vary the number of agents the model is asked to predict in each example. When there are more agents to predict, the task becomes harder because there are many possible good answers, and the model may not predict the exact ground truth. We find that the marginal minADE of the joint prediction model is more sensitive to the number of agents to predict compared to the marginal model. However, for inter-prediction overlap rates (a joint metric we measure) between trajectories, we see that the joint model has overall  lower inter-prediction overlap rates than the marginal model, though overlap rates for both models increase with the number of agents to predict.
>
> 2) In Appendix C “Slicing Results”, we slice by the mean AV speed, the number of predicted agents, and the number of dynamic (moving) agents in the scene, and we include corresponding histograms showing the data distribution across these metadata statistics. Most strikingly, our results show that both the marginal and joint model have poor performance when mean AV speed exceeds 20 m/s. However, looking at the histogram distribution of the data, we also see that there are very few examples where mean AV speed exceeds 20 m/s.  Similarly, when looking at performance as we vary the number of dynamic/moving agents, we see low performance on scenes where the number of moving agents is large, which are also relatively rare in the dataset. Thus, we suspect that the poor performance of the model can be attributed to the lack of training data with the particular metadata statistic.
>
> As suggested by some of the references, metrics such as “Drivable area compliance” / road rule errors that could also be potentially useful unfortunately do not exist on WOMD and would be challenging for us to evaluate manually (e.g., annotating the 40k interactive prediction segments by hand, as was done by prior work such as those you cited, is infeasible), and would probably be its own work.
>
> We’ll also note: The overlap rate on the official WOMD leaderboard just measures prediction to ground truth overlap, not prediction to prediction overlap like one would want to evaluate joint models. We found this unsatisfying, as it doesn’t give a measure of ‘inter-agent consistency’. We went a step further to create an ‘inter-agent consistency’ overlap metric that measures the overlap rate of predictions with each other. We show that joint models do better on this than marginal models, further showing the difference between these two model classes.

---

> > ### Author Response · Authors · 2021-11-16
> > **Response to Reviewer evQV continued**
> >
> > To summarize, we completely agree that the community needs to move towards better joint metrics and evaluation slices of those joint metrics.  As you noted, the community is thinking about improved metrics, but note that the cited work [0] was released (exclusively) on Arxiv in July 2021, months after the Waymo dataset was released; we expect the community will inevitably move towards better metrics and slices.  However, in lieu of the missing metadata from the WOMD dataset, we believe we tried to provide analyses of the interesting slices we could compute.

---

> > > ### Comment · Reviewer_evQV · 2021-11-16
> > > **Response to author comment**
> > >
> > > Thanks for your response.
> > >
> > > I'm glad we agree that the future of the field lies in prediction in interesting and challenging scenarios. I also agree with your stance that analytical comparison in these scenarios is difficult since the field has only just turned in that direction. That is why papers need to at least have a discussion on this topic (not just in supplementary material), even if the discussion is preliminary. **If current papers do not make an active attempt to steer the field, why should one expect the field to move at all?**
> > >
> > > In any case, as it currently stands, the paper's technical novelty is not exciting from a research perspective. Please note that I do *not* dispute the existence of novelty, simply that the contribution made by the paper, given the maturity in the field and the direction this field needs to take, is not of major significance from a research standpoint (maybe from an engineering standpoint).
> > >
> > > Without a discussion that talks about and acknowledges the current efforts (note above references) that is steering the field in the direction it needs to move in, the paper would be more suited towards an autonomous driving workshop, since the major technical contribution is more engineering in nature.
> > >
> > > I would request the authors to consider a discussion section as discussed above in the main paper summarizing their results from appendix B and C and making the motivation of that section clear regarding the current and future efforts of field in challenging and interesting scenarios.
> > >
> > > Cheers,

---

> > > > ### Author Response · Authors · 2021-11-17
> > > > **Response to Response to author comment**
> > > >
> > > > Thank you for the additional feedback!
> > > >
> > > > We would be happy to include a discussion section about this in the main text and include the associated motivating references, something like the following:
> > > >
> > > > "Dataset averages over motion forecasting metrics like min(S)ADE do not capture the types of fine-grained edge cases [citations above] that are critical for measuring and making forward progress in the field.  Unfortunately, the Waymo Motion Open Dataset Interactive Prediction set does not contain pre-defined metadata necessary to evaluate specific interesting slices using different models.  In Appendix X we provide some initial analysis of slices that we can compute from the dataset, which illustrate how joint and marginal model predictions behavior differently as the scene a) scales with more agents in complicated scenes, b) differs in the average speed of agents.  We also provide results on how joint prediction models demonstrate lower inter-prediction overlap compared to marginal models, which we believe to be an additionally important metric to measure going forward."
> > > >
> > > > As mentioned, we do believe we tried to help steer the field by providing this analysis.  To justify our reasoning, we placed this analysis in the appendix believing that the broader ICLR community would care more about the connections between masked language modeling and joint trajectory prediction -- in fact, we think our approach could be relevant to less safety-critical multi-agent prediction scenarios beyond the AV domain.

---

> > > > > ### Comment · Reviewer_evQV · 2021-11-17
> > > > > **Updated evaluation**
> > > > >
> > > > > Thank you for continued discussion. This paragraph looks good and I appreciate your efforts to further this direction. I will update my score based on our discussion. All the best !

---

### Official Review · Reviewer_LaWD · 2021-11-02

**Correctness:** 3
**Technical Novelty And Significance:** 2
**Empirical Novelty And Significance:** 2
**Recommendation:** 5
**Confidence:** 5

**Main Review:**

Section: Abstract, Introduction: Issue with novelty. The manuscript "[combines] a scene-centric approach, agent permutation equivariant model, and a sequence masking strategy...". The first element (specifically, understanding the agent-to-scene interactions, through various attention mechanisms) has been considered, ad nauseum, by several trajectory forecasting works — in the last year alone. The second element is an inherent property of the chosen model class (that remains unsupported/uncited in the text, by the way) and, thus, cannot be used to strengthen the manuscript's contribution. The final element is a well-known practice related, again, to the chosen model architectural class: while it is true that the last element (sequence masking) has not been explored sufficiently in the context of trajectory forecasting (despite the already broad usage, now, of transformer-based architectures), I am hard-pressed to regard this as a strong standalone contribution to the community.

Section: Introduction: The manuscript claims a "unified architecture for prediction and planning". I assert that this not appropriate. The manuscript itself cites Liu et al., (2021a) in reference to the goal-conditioned prediction (GCP) sub-task; however, *that* work is actually performs planning: they actually optimize with respect to a cost formulation, their model includes an inductive bias to learn the transition dynamics of the environment (online), and they evaluate their work in CARLA (an urban driving simulator, for evaluation of online planning and control). In this work, planning-related experiments are not shown: it is another conditional trajectory prediction approach with a transformer-based architecture. I would suggest that this manuscript removes all references to the term 'planning' and claims of the two prediction/planning paradigms being 'unified'.

Section 2: The Related Works section is well-written, save for a couple issues. (1) In most places, the manuscript summarizes the related works and neglects to explain the differences between those works and the present manuscript, let alone explain why the differences are significant. For example, the manuscript states that "LaneGCN (Liang et al., 2020) is agent-centric yet representations are in a global frame – ... the only other work to do so." The manuscript should explain, then, the notable differences that makes the present work uniquely more performant. (2) some concepts and properties are left undefined or unsupported. For example, the manuscript states that "... models that reason in the agent-coordinate frame (Mercat et al., 2020; Zhao et al., 2020; Khandelwal et al., 2020) are intrinsically pose-invariant, but scale linearly with the number of agents, or quadratically with the number of pairwise interactions between agents." Through what mechanism does the present manuscript claim pose-invariance, and what scaling order does the approach in the present manuscript enjoy?

Section 2: "Representing multi-agent futures": The manuscript states that "A common way to represent agent futures is via a weighted set of trajectories per agent... We argue in this work that modeling joint futures in a multi-agent environment (Figure 1, right) is an important concept that has been minimally explored in prior work." One could argue that the implicit weighting for those weighted set of trajectories (for each agent) is a function of other agents’ historical trajectories and the scene context; and, in the case of flow-based approaches, the weighting is also conditioned on a prior density over diverse and admissible trajectories (see Park et al., 2020). Is the manuscript asserting that the implicit weighting, then, does not capture as effectively the same multi-agent/agent-scene interactions as that which is proposed? More discussion is needed to understand why.

Section 3.2: The manuscript attempts to claim as its main novelty the use of masked self-attention for joint prediction in trajectory forecasting problems. If the manuscript wishes to do this, then this inductive bias should be compared with those of other model classes, such as normalizing flow- and CVAE-based approaches. This would provide the community with insights as to why we might choose one over the other; and this work could then claim as a contribution the investigation of these differences, just as [1] did in their discussion of flow (see Figure 9 for comparisons w.r.t RNN-, CVAE-, and GAIL-based configurations).

Section 3.2: "Cross-attention": Not novel. The paper cannot both claim primary novelty from its attention mechanisms and simultaneously ignore similar attention mechanisms as components in other works, as applied to the same tasks. The manuscript should discuss why this "cross-attention" mechanism is better than the plethora of other agent-to-scene attention mechanisms, which have been popular in robot navigation, planning, and trajectory forecasting.

Section 3.2: The manuscript says "Finally, in order to capture both time and agent dependencies, the model simply alternates attention across agents and time in subsequent layers (Figure 2, right panel)." Why interleave? Why not calculate both attention distributions — across both agent and time dimensions — serially?

Section 3.4: The referenced 'Figure 7' appears to be missing.

Section 4.4: The manuscript asserts that "...the multi-task model matches the performance of our MP-only trained joint model on both joint (Table 3) and marginal metrics (see also Appendix, Table 10), indicating that the addition of tasks does not degrade standard motion prediction performance." This claim is not really apparent from the provided results: throughout the tables, the performance of the multi-task models are always subpar or inconclusive, especially since no standard error calculations are given to contextualize the numerical results.

Sections 4, B: Results of the various ablations are very sensitive to the model instance's training objective (marginal, joint, multi-task) and the trajectory lengths (t={3, 5, 8}). To me, this renders such claims as "we [produce] consistent futures that account for interactions between agents" (Abstract, Intro) somewhat inconclusive, still.

Section 4: The manuscript states: "Although no quantitative benchmarks are available in the community for quantifying GCP predictions, we take these results as positive indication that the model is responding appropriately and save further exploration of counterfactual analysis for future work." Consider evaluation in online urban-driving settings, like the CARLA benchmarks: Original, NoCrash, Anyweather, CARNOVEL [2], or CARLA Challenge. See also Liu et al., (2021a), which this manuscript even cites.

----

[1] Rhinehart, Nicholas, Kris M. Kitani, and Paul Vernaza. "R2p2: A reparameterized pushforward policy for diverse, precise generative path forecasting." In Proceedings of the European Conference on Computer Vision (ECCV), pp. 772-788. 2018.

[2] Filos, Angelos, Panagiotis Tigkas, Rowan McAllister, Nicholas Rhinehart, Sergey Levine, and Yarin Gal. "Can autonomous vehicles identify, recover from, and adapt to distribution shifts?." In International Conference on Machine Learning, pp. 3145-3153. PMLR, 2020.

**Summary Of The Paper:**

The manuscript proposes the use of a factorized attention mechanism and a masked transformer-based architecture, for conditional trajectory forecasting. Experiments on Argoverse and the Waymo Open Motion Dataset are provided.

**Summary Of The Review:**

I have some issues with the novelty of the work presented, the accuracy of the claims made, and the somewhat lacking comparison with the related work and key ablation experiments. See main review, above.

---

> ### Author Response · Authors · 2021-11-16
> **Response to Reviewer LaWD**
>
> We thank the reviewer for their comments, and respond to each of their concerns below.
>
> **Novelty**: One of the strengths of our approach is how consistent it is – the ability for transformers to be effective for both modeling agent and roadgraph interactions provides the insight that simpler models can be effective for this task, instead of more bespoke combinations. This stems from the connection between masked language modeling to trajectory prediction, which we appreciate your acknowledgement that is novel. We think this paper opens up a new direction of thinking that can help further cross pollinate ideas between these research communities.
>
> **Claims on planning**: We agree with your assertion, and also highlight that this submitted manuscript does not claim that we have a “unified architecture for prediction and planning”. The closest wording we used was “The goal of this work is to take a step in the direction of unifying motion prediction and planning by developing a model that can exploit conditioning information, such as the AV’s goal, and produce joint consistent predictions about the future for all agents simultaneously.” We believe that our flexible trajectory prediction model can be useful for planning, but do not claim that it performs planning. We would be happy to tune the language to make this clearer in the submission.
>
> **Related Works**: Similarly, we never made the claims that “implicit weighting does not capture multi-agent interactions”. We strive to represent the prior work accurately as we can, and would be happy to make the necessary adjustments here to reflect that further.
>
> **Permutation Equivariance**: Our choice of model architecture is permutation equivariant as all the layers that form the model are also permutation equivariant (with respect to agent ordering). While there are many approaches to forming a permutation equivariant architecture, our method is simple and uses transformers consistently throughout. While each ingredient in the architecture has been proposed before, our recipe of putting it altogether in a parsimonious and effective way is new. We will add more supporting details on the permutation equivariance properties of the model in the paper.
>
> **Model Details (3.2)**: We would be happy to add references of other works which also use cross attention on the same task, and a discussion on why cross-attention is useful here. In terms of interleaving vs serial arrangement: interleaving allows the model to better propagate information across time and agents. Conceptually, we think of two adjacent layers (one time, one agents) as a more effective version than a single layer that just performed attention across both axes simultaneously. We also provide an ablation on this specifically, showing quality improvements with the factorized attention.
>
> **Results (4.4)**: We agree that standard error calculations would be beneficial here. However, the differences are small enough that we think goes to support the fact that the multi-task model does not suffer the usual trade-offs where one task clearly performs much worse. As suggested, we would work to soften our language used here since we do not have the necessary error bars to make a statistically significant claim.
>
> **Section 4**: Thank you for the references to the CARLA benchmarks.

---

### Official Review · Reviewer_ADF1 · 2021-11-03

**Correctness:** 4
**Technical Novelty And Significance:** 4
**Empirical Novelty And Significance:** 3
**Recommendation:** 8
**Confidence:** 4

**Main Review:**

Strengths:
1. The scene transformer has its flexibility to switch between different prediction tasks by using different masking strategies.
2. The model has proved to generate competitive results on both marginal and joint predictions compared to recent SOT algorithms.

Weakness:
Since model flexibility using the mask is the highlighted benefit of this model, I was expecting more experiments and discussion for the other prediction modes, such as conditional motion prediction and goal conditioned prediction. They are more interesting part of the paper that may lead a new direction for prediction model. However, the current result are still too preliminary. It would be better to provide more quantitative results to justify the advantage of such a unified model on CMP or GCP.

Several design choices on components are described, such as factorized self attention and cross attention. An ablation study to summarize these findings would be better to review for the readers.

**Summary Of The Paper:**

The paper presents a unified architecture to predict future motion trajectories of all agents in the scene, while interactions between agents are captured. A Transformer based architecture is designed. It also uses a masking strategy as a query, enabling one to condition on hypothetical agent futures during the inference time, and achieves conditional motion prediction or goal conditioned prediction. The model was tested on two standard datasets, and achieves state of the art performance.

**Summary Of The Review:**

Overall this is a strong paper with sufficient novelty in its model architecture in the context of motion prediction for autonomous driving. The experiment results on standard prediction tasks on popular benchmarks are convincing. Despite some minor issues in presentation, the paper is above the acceptance bar.

---

> ### Author Response · Authors · 2021-11-16
> **Response to Reviewer ADF1**
>
> We thank the reviewer for their comments and appreciate that they find the architecture novel with convincing results.
>
> We will look into adding more ablation results. For the unified model on CMP and GCP, one result that is already present on the GCP task is in the Appendix (see Figure 6). We also have an ablation study on the factorized self-attention module (see Section 4.3) where we find that the factorized version out-performed the non-factorized version.

---

### Decision · Program_Chairs · 2022-01-20

**Decision:**

Accept (Poster)

**Comment:**

The paper shows interesting and discussion inspiring results on multi-agent trajectory prediction, as needed, for instance, in autonomous driving. Among the key technical ideas is a “conditional scene transformer” approach for flexible predictions for different agents.  Results on two  public benchmarks are impressive. Some reviewers are a bit torn about the significance of the technical contributions and the analyses of the results. Nevertheless, on average, the reviewers vote the paper to be above the acceptance threshold.